# BTL-UI: Blink-Think-Link Reasoning Model for GUI Agent

**Shaojie Zhang**[*]  **Ruoceng Zhang**[*]  **Pei Fu**[*]  **Shaokang Wang**  **Jiahui Yang**
**Xin Du**  **Shiqi Cui**  **Bin Qin**  **Ying Huang**  **Zhenbo Luo**[†]  **Jian Luan**

MiLM Plus, Xiaomi Inc
{zhangshaojie5, zhangruoceng1, fupei1, luozhenbo, luanjian}@xiaomi.com

## Abstract

In the field of AI-driven human-GUI interaction automation, while rapid advances in multimodal large language models and reinforcement fine-tuning techniques have yielded remarkable progress, a fundamental challenge persists: their interaction logic significantly deviates from natural human-GUI communication patterns. To address this gap, we propose Blink–Think–Link (BTL), a brain-inspired framework for human-GUI interaction that mimics the human cognitive process between users and graphical interfaces. The system decomposes interactions into three biologically plausible phases: (1) **Blink** - rapid detection and attention to relevant screen areas, analogous to saccadic eye movements; (2) **Think** - higher-level reasoning and decision-making, mirroring cognitive planning; and (3) **Link** - generation of executable commands for precise motor control, emulating human action selection mechanisms. Additionally, we introduce two key technical innovations for BTL framework: (1) Blink Data Generation - an automated annotation pipeline specifically optimized for blink data, and (2) BTL Reward – the first rule-based reward mechanism that enables reinforcement learning driven by both process and outcome. Building upon this framework, we develop a GUI agent model named BTL-UI, which demonstrates competitive performance across both static GUI understanding and dynamic interaction tasks in comprehensive benchmarks. These results provide conclusive empirical validation of the framework's efficacy in developing advanced GUI agents.

## 1  Introduction

Automation of graphical user interface (GUI) interactions constitutes a pivotal milestone in developing genuinely intelligent digital assistants [1, 2, 3]. Recent breakthroughs in large vision-language models (VLMs) [4, 5] and reinforcement learning fine-tuning techniques have substantially improved agents' capabilities in natural language command interpretation, visual element perception, and multi-step task execution through human-like reasoning [6, 7].

However, current mainstream systems adopt mainly two approaches. The first relies on supervised fine-tuning (SFT) to align model behavior with task objectives, but this method faces two major limitations: a strong dependence on large-scale expert-labeled data and limited generalization capability when faced with out-of-distribution scenarios. The second approach involves rule-based reinforcement fine-tuning (RFT) [8], as shown in Figure 1 (a), which enhances generalization in complex tasks by using a structured reasoning format: intermediate cognitive steps are encapsulated within *<think>* tags, and final decisions are expressed through *<answer>* tags. Although effective in improving task performance, these methods [9, 10, 11] exhibit two critical shortcomings: (1)

---

[*] Equal contribution; [†] Corresponding author.
[§] https://github.com/xiaomi-research/btl-ui

39th Conference on Neural Information Processing Systems (NeurIPS 2025).

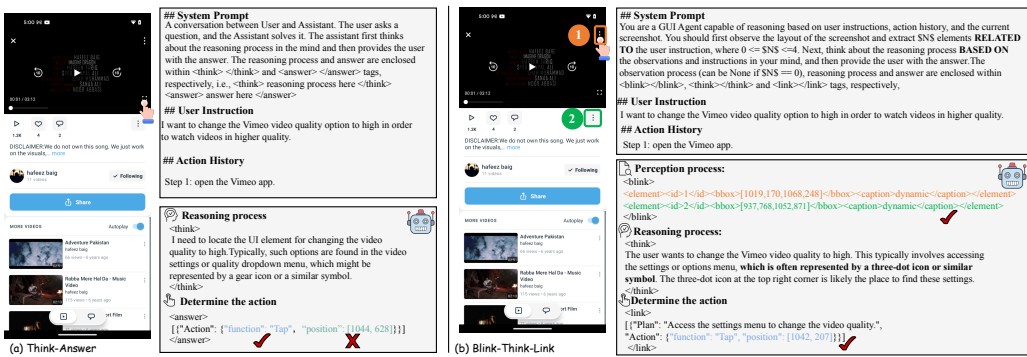

Figure 1: Framework comparison of previous Think-Answer and Blink-Think-Link in GUI tasks for RFT. Specifically, colorful text is supervised by rule-based reinforcement learning. And different colors of text indicate different reward rules. The previous "Think-Answer" framework is optimized by format reward, action type reward, and corresponding args reward. And our Blink-Think-Link framework is optimized by dual format reward, blink reward, and link reward.

significant deviation from natural human-GUI interaction patterns, and (2) excessive focus on interaction outcomes while lacking effective process-oriented reward mechanisms.

Cognitive studies [12, 13, 14] demonstrate that human-GUI interaction achieves remarkable efficiency through three sequential processes: (a) **Blink Phase**. Rapid target location during saccadic intervals; (b) **Think Phase**. Multimodal information integration with intentional reasoning; (c) **Link Phase**. Generation of precise motor execution commands. Building upon this cognitive finding, we innovatively propose a biologically inspired interaction paradigm—the **B**link-**T**hink-**L**ink (**BTL**) paradigm—for GUI agents, and computationally simulate this paradigm through a structured output mechanism (as shown in Figure 1 (b)):

- **<blink>**: Where relevant areas of the screen are rapidly located, analogous to saccadic eye movements. The visual attention-related region-of-interest information is encapsulated within *<blink></blink>* tags.

- **<think>**: Where the system engages in high-level reasoning and decision-making, mirroring cognitive task planning. The reasoning processes are recorded in *<think></think>* tags.

- **<link>**: Where actionable commands are generated for precise execution, reflecting human action selection mechanisms. The action commands are output in *<link></link>* tags.

Specifically, to model human visual localization capabilities during blink intervals, we developed an innovative blink data generation pipeline to automatically produce several region-of-interest (ROI) annotations for training samples. Furthermore, to address the limitations of current reward models in rule-based RFT algorithms that over-rely on outcome-based rewards while neglecting guidance for intermediate interaction processes, We propose the innovative **BTL Reward**, a Process-Outcome Integrated Reward Mechanism, which comprises three core components: (1) the Dual Format Reward for template and content matching, (2) the Blink Reward for fine-grained guidance of interaction processes, and (3) the Link Reward for action outcome evaluation. By combining the Blink Reward's granular process supervision with the Link Reward's precise outcome feedback, this mechanism pioneers the organic integration of process-oriented and outcome-driven approaches. Compared to conventional reward schemes focusing solely on final outcomes, the BTL reward mechanism delivers more sophisticated and multi-dimensional training guidance. Finally, building upon this framework, we develop BTL-UI, a GUI Agent that demonstrates the framework's effectiveness across multiple GUI tasks.

In general, the main contributions are summarized as follows:

1. We propose BTL (Blink-Think-Link), an innovative framework that simulates the human cognitive process in the human-GUI interaction by explicitly modeling how users perceive, process, and act upon interface elements.

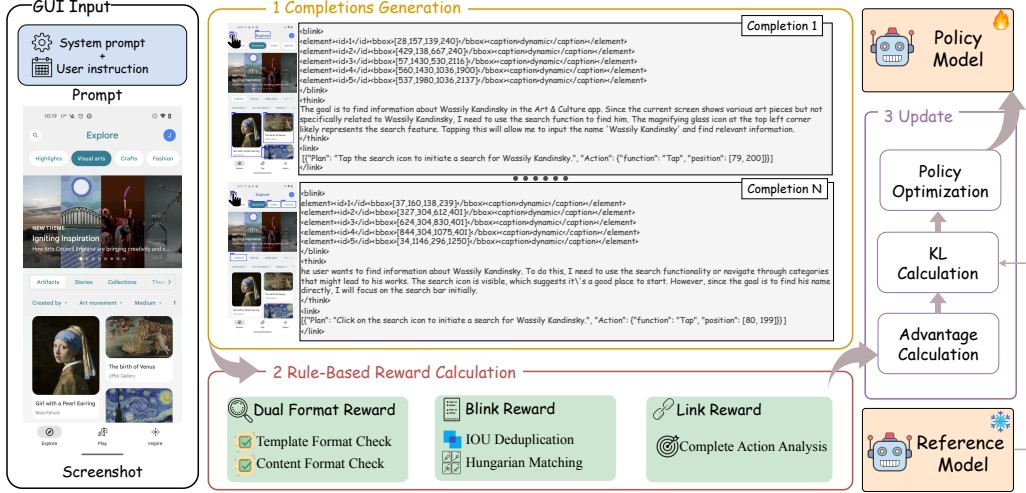

Figure 2: Overall framework of BTL. We adopt Group Relative Policy Optimization (GRPO) to optimize the proposed BTL. Firstly, the base model generates $N$ completions for a given GUI task sample. Furthermore, GRPO computes the relative advantages within a group of completions, eliminating the need for manually annotated data. Finally, the policy model updates parameters under the guidance of relative advantages and the KL divergence constraint.

2. We propose two key innovations to jointly advance the learning of GUI agents within this framework: (1) Blink Data Generation—an efficient data annotation pipeline that automatically generates multi-region Regions of Interest (ROIs) for training samples; (2) BTL Reward—the first rule-based Process-Outcome Integrated Reward Mechanism.

3. We develop BTL-UI, a GUI agent trained via the BTL framework, and extensive experiment results demonstrate that the model achieves competitive performance across multiple GUI benchmarks.

## 2 Related Work

### 2.1 GUI Agents

Autonomous agents powered by large language models (LLMs) and VLMs have recently garnered considerable scholarly interest due to their interactive functionalities. For GUI tasks, earlier systems relied on LLMs to read and interpret structured representations such as HTML and accessibility trees. However, since icons, images, diagrams, and their spatial relationships are difficult to express in such structured languages, agents based on LLM often perform poorly [6, 15, 16]. Therefore, VLM-based agents have been introduced to perceive visual GUI signals directly with better performance [2, 17, 18, 19, 20, 21]. For example, UGround [22] developed a specialized GUI grounding model for GUI element localization. OS-Atlas [23] proposed a foundational model for GUI agents by interpreting human intentions and predicting actions in the form of function calls. Aguvis [24] integrated explicit planning and reasoning within the model, enhancing its ability to navigate and interact with complex digital environments autonomously. UI-TARS [9] combines GUI-related pretraining with task-level reasoning fine-tuning to better capture the complexity of GUI interactions. Although research on VLM-based GUI agents has made impressive progress, they mainly follow the SFT training paradigm, which directly mimics the ground-truth actions provided in the curated data.

### 2.2 Reinforcement Fine-Tuning

With the advent of rule-based reinforcement learning approaches such as OpenAI-o1 [25] and DeepSeek-R1 [8], recent studies have demonstrated that RFT improves the reasoning abilities of the model and provides greater generalizability [26]. Subsequent approaches [26, 27, 28] have introduced this paradigm to VLMs. For example, Vision-R1 [27] combined a vision criterion-driven reward

function and a progressive rule refinement strategy to enhance VLM's object localization capabilities. Visual-RFT [26] adopted the reinforcement learning strategy to enhance visual perception and grounding ability of VLMs. VLM-R1 [29] demonstrated that RFT with small amounts of high-quality data can enable VLMs to solve complex vision-language tasks.

For GUI tasks, UI-R1 [30] and GUI-R1 [10] introduced rule-based reinforcement learning frameworks that require minimal expert supervision, demonstrating competitive performance. InfiGUI-R1 [11] further advanced the field by bridging reactive execution and deliberative reasoning through the Actor2Reasoner architecture. However, existing RFT-based GUI agents predominantly adopt rule-based reinforcement learning, which focus on final outcomes and lack intermediate process guidance, often overlooking key aspects of human cognition and interaction.

## 3  Method

In this section, we introduce BTL, a new framework grounded in cognitive science theory, with its core concept derived from the Blink-Think-Link paradigm observed in human-GUI interactions. The framework is shown in Figure 2. We detail the implementation details of this framework through the following components: Preliminaries, Blink Data Generation, BTL Reward, and Policy Optimization.

### 3.1  Preliminaries

The interaction between a GUI agent and its environment can be naturally formulated as a Markov Decision Process (MDP), defined by the tuple $\langle \mathcal{S}, \mathcal{A}, \mathcal{Z}, \mathcal{T}, \mathcal{O} \rangle$. Here, $\mathcal{S}$ denotes the state space representing possible screen states; $\mathcal{A}$ is the action space that encompasses interaction types such as clicking, typing, and scrolling; $\mathcal{Z}$ is the observation space, including screenshots or structured UI representations; $\mathcal{T} : \mathcal{S} \times \mathcal{A} \times \mathcal{S} \to [0, 1]$ defines the probability of transitioning from one state to another given an action; and $\mathcal{O} : \mathcal{S} \times \mathcal{A} \to \mathcal{Z}$ specifies the probability of receiving a particular observation given a state and an action.

During task execution, at each discrete time step $t$, the agent receives an input tuple $(z_t, u, h)$, where $z_t \in \mathcal{Z}$ is the current state of the screen, $u$ refers to the global task instruction and $h$ is its interaction history. The BTL process can then be formalized as a structured policy function $\mathcal{F}$:

$$\mathcal{F}(\{z_t, u, h\}) \to o_t = \{b_t, d_t, a_t\}, \tag{1}$$

where $o_t$ denotes the BTL output at time $t$, consisting of: $b_t$—visual attention regions, $d_t$—reasoning and decision trace, $a_t$—the final action to be executed. Each action $a_t = (\alpha_t, \delta_t) \in \mathcal{A}$ is composed of an action type $\alpha_t$ (e.g., click) and its corresponding parameters $\delta_t$ (e.g., coordinates, text input). Upon execution of $a_t$, the environment transitions to a new state $z_{t+1}$, and the process repeats until the task is completed or a terminal condition is met.

### 3.2  Blink Data Generation

One of the core innovations of the BTL framework is its ability to simulate the human mechanism of rapidly locating ROIs in a visual scene during the blink phase. To achieve this, we propose an automated Blink data generation pipeline that annotates ROIs on the screenshot corresponding to the user instruction in the MDP. As illustrated in Figure 3, the pipeline consists of two main stages. A parsing model [31] first processes the raw screenshot to extract semantic UI elements. Then, an analysis model [5] is used to evaluate the visual importance and contextual relevance of these elements, allowing filtering and prioritization to produce the final ROI annotations.

Specifically, in the first stage, we extract individual UI elements such as buttons, icons, and text fields, annotating each with bounding box coordinates, type, and semantic captions. These annotations form a structured representation of the state of the screen, enabling bottom-up human-like interpretation of the GUI. The output of this stage is a comprehensive list of elements, denoted $\mathcal{E} = \{e_1, e_2, \ldots, e_n\}$, where each $e_k = \{\mathrm{id}_k, \mathrm{bbox}_k, \mathrm{type}_k, \mathrm{caption}_k, \mathrm{interactivity}_k\}$ represents the attributes related to the element. This foundational representation serves as the input for subsequent filtering and prioritization steps that model instruction-directed visual attention.

In the second stage, we employ Qwen2.5-VL-32B [5] to simulate top-down attention by filtering and ranking elements based on visual saliency and task relevance. Oriented by task instruction $u$ and

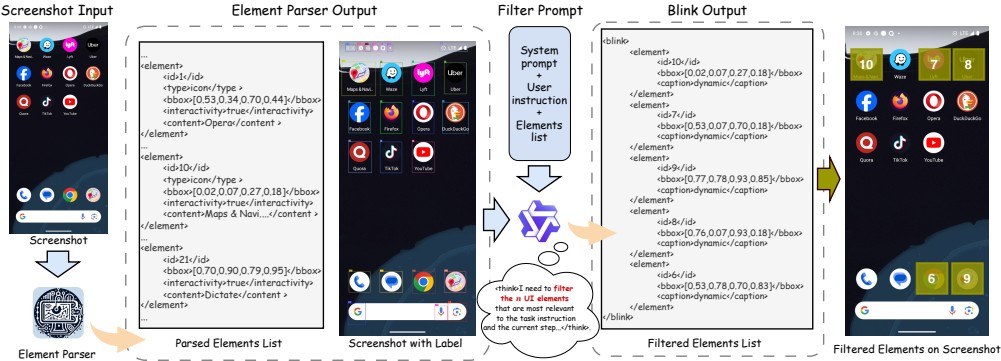

Figure 3: Two-stage data construction pipeline. In the first stage, the basic properties of UI elements are obtained by a parsing model. To eliminate the redundancy of the number and attributes of elements, the analysis model in the second stage simplifies the list to $\lambda$ elements with their positions (*<bbox>*), while the reserved *<caption>* attribute indicates whether the element is interactive. In the example shown in the figure, the instruction for the current step is "Use the GPS to locate a nearby museum and then book a ride with Lyft." Accordingly, the most relevant element in the Blink output is the "Maps & Navigation" app with *<ID>10</ID>*.

interaction history $h$, the model dynamically selects a subset of elements from the filtered list $\mathcal{E}$. This selection process can be formulated as:

$$\mathcal{E}_{\text{ROI}} = f(\mathcal{E}, u, h), \tag{2}$$

where $\mathcal{E}_{\text{ROI}} \subseteq \mathcal{E}$ denotes the resulting set of $\lambda$ candidate elements that are most relevant to the current task step during blink phases, encapsulated within *<blink></blink>* tags. The choice of the number of elements after filtering, $\lambda$, achieves a trade-off between BTL performance and efficiency. This attention-guided annotation not only mimics human visual focus during blink phases but also provides a high-quality reference for optimizing the agent policy.

## 3.3 BTL Reward

Effective GUI agents must excel at both interface grounding and long-horizon planning. To this end, we design a three-component rule-based reward scheme, denoted as $R_{\text{BTL}}$ that mirrors the human Blink–Think–Link cognitive cycle:

$$R_{\text{BTL}} = R_{\text{format}} + R_{\text{blink}} + R_{\text{link}}. \tag{3}$$

Each term provides targeted supervision at a different phase of interaction, as detailed below.

**Dual Format Reward.** Following previous work [8, 26] that leverages format rewards to encourage predefined templates for easy answer extraction, we introduce a dual format reward to evaluate whether the generated output adheres to both the expected structural template and content. Specifically, the template check function $f_{\text{template}}$ is used to check whether the generated completions meet the Blink-Think-Link three-stage grammatical structure. Furthermore, the content check function $f_{\text{content}}$ is adopted to evaluate whether the blink content complies with the XML format and the link content complies with the JSON format, which facilitates the parsing of trajectory planning and actions with corresponding arguments. We adopt a binary reward scheme, assigning a reward of 1 only when the prediction $o_i$ fully satisfies both format and content criteria as follows:

$$R_{\text{format}}(o_i) = \begin{cases} 1, & \text{if } f_{\text{template}} = 1 \wedge f_{\text{content}} = 1 \\ 0. & \text{otherwise} \end{cases} \tag{4}$$

**Blink Reward.** This component incentivizes the rapid and accurate localization of the interface elements relevant to the instruction $u$. From the agent's prediction $o_i$, we extract a set of ROIs $P_i = \{p_i^x\}$ and compare them to ground-truth annotations $G_i = \{g_i^x\}$ (see §3.2). We adopt the Hungarian matcher [32] $M(\cdot, \cdot, \tau)$, a classical assignment algorithm used to compute the optimal

one-to-one matching between predicted and ground-truth bounding boxes based on IoU scores, under a given threshold $\tau$.

$$index_{\text{match}} = \big\{ y \mid g_i^y \in G_i, \ \exists p_i^x \in P_i \ \text{s.t.} \ M(p_i^x, g_i^y, \tau) = 1 \big\}. \tag{5}$$

It is worth noting that in the planning task, the elements related to the instruction $u$ may not be explicitly present in the current screenshot. And the corresponding operation should be to other pages through scrolling or going back. Thus, in the predicted results, $P_i = \varnothing$ is allowed. Consequently, the blink reward can be defined as follows:

$$R_{\text{blink}}(o_i) = \begin{cases} 1, & \text{if } P_i = \varnothing \wedge \big(G_i = \varnothing \vee a_i \in \mathcal{A}^*\big) \\ \max \big\{ s(y) \mid y \in index_{\text{match}} \big\}, & \text{elif } P_i \neq \varnothing \wedge G_i \neq \varnothing \\ 0. & \text{otherwise} \end{cases} \tag{6}$$

where $\mathcal{A}^*$ denotes the non-interactive action spaces and $s(\cdot)$ refers to the reward allocation function, which is determined based on the priorities of elements in the annotations.

**Link Reward.** The link phase assesses the agent's ability to generate a fully coherent executable command. Recent RFT-based GUI agents [10, 11, 30] always split the reward of the predicted action into a reward for the action type and a reward for the action args (e.g. click coordinates or input text). However, this kind of reward will split an action into two independent contents, which is not in line with human cognition. At the same time, this staged action reward will cause reward hacking, which prevents the agent from understanding the designed action space. Thus, we employ a strict binary criterion: the agent receives a reward only if both the action type and its associated arguments are exactly correct. Formally, the link reward is defined as:

$$R_{\text{link}}(o_i) = \begin{cases} 1, & \text{if } f_{\text{type}} = 1 \wedge f_{\text{args}} = 1 \\ 0. & \text{otherwise} \end{cases} \tag{7}$$

This all-or-nothing scheme ensures that the final command is internally consistent and accurately reflects the intended GUI operation.

## 3.4 Advantage Computation and Parameter Update

As shown in Figure 2, we adopt Group Relative Policy Optimization (GRPO) to optimize the proposed BTL. Since its supervision is based solely on the final outcome, GRPO is particularly suited for tasks with explicit, objective answers. Furthermore, GRPO significantly reduces memory overhead for VLMs by removing the reward models or value models in other performance optimization methods [33].

Given a base model to be optimized, GRPO starts by initializing a trainable policy model $\pi_\theta$ and a frozen reference model $\pi_{\text{ref}}$. For a given GUI task sample $\{z_i, u, h\}$, the policy model $\pi_\theta$ first generates a group of completions $\{o_1, o_2, ..., o_N\}$. Then, the reward function computes the whole group's rewards $\{R_1, R_2, ..., R_N\}$, which are further used to calculate the advantage $A_i$ of each completion within the group by:

$$A_i = \frac{R_i - \text{mean}(\{R_j\}_{j=1}^N)}{\text{std}(\{R_j\}_{j=1}^N)}. \tag{8}$$

After the reference model computes the logits to output each completion given the task, the policy model $\pi_\theta$ is optimized by maximizing the following objective:

$$\mathcal{J}_{GRPO}(\theta) = \frac{1}{N} \sum_{i=1}^N \left( \frac{\pi_\theta(o_i|z_i, u, h)}{\pi_{\theta_{\text{old}}}(o_i|z_i, u, h)} A_i - \beta \cdot \mathcal{KL}(\pi_\theta(o_i|z_i, u, h) | \pi_{\text{ref}}(o_i|z_i, u, h)) \right), \tag{9}$$

where $N$ is the number of completions in one group and $\beta$ is the hyperparameter to control the KL divergence constraints. This objective motivates the model to tend to produce the completion with a higher advantage within a group, but not to stray too far from the initial model.

## 4 Experiment

### 4.1 Implementation Details

**Experimental Setup.** We develop the BTL-UI-3B/7B model based on Qwen2.5-VL-3B/7B and adopt the ms-swift framework [34] for RL training. As shown in Table 1, we train BTL-UI in a mix of grounding and planning data.

**Evaluation.** To conduct a thorough evaluation of BTL-UI, we employ a range of critical benchmarks that focus on specific aspects of the GUI agent's grounding and planning capabilities:

*Grounding*: Screenspot series benchmarks assess fundamental GUI understanding and element grounding accuracy across diverse platforms (Mobile, Desktop, Web). ScreenSpot [7] evaluates the single-step GUI grounding performance across multiple platforms. ScreenSpot-V2 [7], a re-annotated

Table 1: RFT data for BTL-UI.

| Category | Source | Size |
|---|---|---|
| Grounding | ShowUI-Web [35] | 1K |
| | ShowUI-Desktop [35] | 1K |
| Low-Level | AndroidControl [36] | 500 |
| | GUI-Odyssey [37] | 500 |
| High-Level | AndroidControl [36] | 500 |
| | GUI-Odyssey [37] | 500 |

version, addresses annotation errors present in the original ScreenSpot. ScreenSpot-Pro [38] specifically increases the difficulty with complex desktop applications and high-resolution screens.

Table 2: GUI grounding accuracy on ScreenSpot [7]. **Bold** means the best results, and underline means the second best results. Avg. denotes the average performance on mobile, desktop and web subtasks.

| Model | Method | Model Size | Mobile | | Desktop | | Web | | Avg. |
|---|---|---|---|---|---|---|---|---|---|
| | | | Text | Icon | Text | Icon | Text | Icon | |
| GPT-4o [39] | ZS | - | 30.5 | 23.2 | 20.6 | 19.4 | 11.1 | 7.8 | 18.8 |
| Qwen2-Vl [4] | ZS | 7B | 75.5 | 60.7 | 76.3 | 54.3 | 35.2 | 25.7 | 55.3 |
| OS-Atlas-Base [23] | ZS | 7B | 93.0 | 72.9 | 91.8 | 62.9 | 90.9 | 74.3 | 82.5 |
| Qwen2.5-VL [5] | ZS | 3B | 90.5 | 61.1 | 60.0 | 43.2 | 80.9 | 40.0 | 65.0 |
| Qwen2.5-VL [5] | ZS | 7B | 86.3 | 83.8 | 85.6 | 67.1 | 87.4 | 78.6 | 84.8 |
| InternVL3 [40] | ZS | 8B | - | - | - | - | - | - | 79.5 |
| CogAgent [6] | SFT | 18B | 67.0 | 24.0 | 74.2 | 20.0 | 70.4 | 28.6 | 47.4 |
| Aria-UI [41] | SFT | 3.9B | 92.3 | 73.8 | 93.3 | 64.3 | 86.5 | 76.2 | 82.4 |
| SeeClick [7] | SFT | 9.6B | 78.0 | 52.0 | 72.2 | 30.0 | 55.7 | 32.5 | 53.4 |
| ShowUI [35] | SFT | 2B | 92.3 | 75.5 | 76.3 | 61.1 | 81.7 | 63.6 | 75.1 |
| Aguvis [24] | SFT | 7B | 95.6 | 77.7 | 93.8 | 67.1 | 88.3 | 75.2 | 84.4 |
| UGround [22] | SFT | 7B | 82.8 | 60.3 | 82.5 | 63.6 | 80.4 | 70.4 | 73.3 |
| UGround-V1 [22] | SFT | 2B | 89.4 | 72.0 | 88.7 | 65.7 | 81.3 | 68.9 | 77.7 |
| UGround-V1 [22] | SFT | 7B | 94.1 | 79.9 | 93.8 | 76.4 | 90.9 | 84.0 | 86.3 |
| UI-TARS [9] | SFT | 2B | 93.0 | 75.5 | 90.7 | 68.6 | 84.3 | 74.8 | 82.3 |
| UI-TARS [9] | SFT | 7B | 94.5 | **85.2** | **95.9** | **85.7** | 90.0 | 83.5 | **89.5** |
| UI-R1 [30] | RFT | 3B | - | - | 90.2 | 59.3 | 85.2 | 73.3 | - |
| GUI-R1 [10] | RFT | 3B | - | - | 93.8 | 64.8 | 89.6 | 72.1 | - |
| GUI-R1 [10] | RFT | 7B | - | - | 91.8 | 73.6 | **91.3** | 75.7 | - |
| **BTL-UI** | RFT | 3B | 96.3 | 77.3 | 88.2 | 57.9 | 80.0 | 68.9 | 80.0 |
| **BTL-UI** | RFT | 7B | **97.1** | 83.8 | 90.2 | 70.7 | 88.7 | **84.5** | 87.2 |

*Planning*: AndroidControl [36] and GUI-Odyssey [37] evaluate the agent's grounding and planning ability to execute multi-step tasks within realistic Android environments. These benchmarks provide agents with a task instruction, a current screenshot, and previous interaction history, aimed at enabling accurate prediction of the next action. Furthermore, according to the input, the settings on AndroidControl can be divided into low-level tasks and high-level tasks. High-level tasks only input the global instruction to the agent, while low-level tasks will additionally input the single-step action plan. And GUI-Odyssey only adopts the high-level experimental setups.

**Evaluation Metrics.** For grounding tasks, we use click point prediction accuracy as our evaluation metric. For planning tasks, according to OS-Atlas [23], we report three standard metrics for GUI agents: action type prediction accuracy (Type), click point prediction accuracy (GR) and step success rate (SR). Specifically: **Type** measures the exact-match accuracy between predicted and ground-truth

Table 3: GUI grounding accuracy on ScreenSpot-V2 [7]. **Bold** means the best results, and underline means the second best results. Avg. denotes the average performance on mobile, desktop and web subtasks.

| Model | Method | Model Size | Mobile | | Desktop | | Web | | Avg. |
|---|---|---|---|---|---|---|---|---|---|
| | | | Text | Icon | Text | Icon | Text | Icon | |
| GPT-4o [39] | ZS | - | 30.5 | 23.2 | 20.6 | 19.4 | 11.1 | 7.8 | 18.8 |
| OS-Atlas-Base [23] | ZS | 4B | 85.7 | 58.5 | 72.2 | 45.7 | 82.6 | 63.1 | 70.1 |
| OS-Atlas-Base [23] | ZS | 7B | 93.0 | 72.9 | 91.8 | 62.9 | 90.9 | 74.3 | 82.5 |
| Qwen2.5-VL [5] | ZS | 3B | 92.1 | 66.8 | 72.6 | 46.8 | 83.0 | 44.3 | 70.4 |
| Qwen2.5-VL [5] | ZS | 7B | 97.9 | 86.7 | 87.6 | 68.6 | 91.5 | 79.3 | 87.1 |
| InternVL3 [40] | ZS | 8B | - | - | - | - | - | - | 81.4 |
| SeeClick [7] | SFT | 9.6B | 78.4 | 50.7 | 70.1 | 29.3 | 55.2 | 32.5 | 55.1 |
| Aguvis [24] | SFT | 7B | 95.6 | 77.7 | 93.8 | 67.1 | 88.3 | 75.2 | 84.4 |
| UI-TARS [9] | SFT | 2B | 95.2 | 79.1 | 90.7 | 68.6 | 87.2 | 78.3 | 84.7 |
| UI-TARS [9] | SFT | 7B | 96.9 | 89.1 | **95.4** | **85.0** | 93.6 | **85.2** | **91.6** |
| **BTL-UI** | RFT | 3B | 97.9 | 83.4 | 88.7 | 62.1 | 83.3 | 69.0 | 82.9 |
| **BTL-UI** | RFT | 7B | **98.6** | **89.6** | 92.3 | 70.7 | 92.3 | 80.3 | 89.1 |

action types (e.g., "click" vs. "swipe"). **GR** evaluates grounding performance via click point prediction accuracy in specific action types (e.g. "click" and "long press"). **SR** is the step-wise success rate: a step is counted as successful only if both the predicted action and its associated arguments (e.g., click coordinates or input text) match the ground truth.

## 4.2 Experimental Results

We evaluate BTL-UI across three key capabilities: grounding, low-level planning, and high-level reasoning. The results demonstrate consistent and significant improvements over existing baselines, validating the effectiveness of the Blink-Think-Link framework.

**Grounding Capability.** To assess how well the model can localize UI elements, we report grounding accuracy on the ScreenSpot benchmark series in Table 2, 3, and 4. In the original ScreenSpot dataset, BTL-UI-7B achieves an average accuracy of 87.2%, outperforming the baseline Qwen2.5-VL-7B (84.8%) and surpassing the supervised fine-tuned Aria-UI (82.4%). On the corrected ScreenSpot-V2, performance further improves to 89.1%. In the ScreenSpot-Pro benchmark, the BTL-UI-3B consistently outperforms other RFT-based models, achieving an average accuracy of 27.1%, which is substantially higher than UI-R1 (17.8%) on the same scale. This suggests that the Blink Phase, which encourages early-stage attention to semantically relevant regions through ROI supervision, enables more precise perception and grounding even under diverse visual layouts. Although the overall grounding performance of BTL-UI remains slightly lower than that of UI-TARS [9], which is a strong GUI Agent developed based on Qwen2-VL [4] with training on 50B tokens, the proposed BTL-UI shows certain advantages in the mobile subtasks.

**Planning Capability.** As shown in Table 5, BTL-UI exhibits strong generalization and reasoning ability on both low-level and high-level GUI planning benchmarks. In AndroidControl-Low, BTL-UI-7B achieves an SR of 88.0%, surpassing the previous best model OS-Atlas-Pro-7B (85.2%) and GUI-R1-7B (66.5%) , while the 3B variant attains a comparable 84.8%, confirming the efficiency of the BTL reinforcement paradigm. For long-horizon tasks in AndroidControl-High, which require multi-step reasoning and contextual grounding, BTL-UI-7B achieves an SR of 69.2%, outperforming GUI-R1-7B (51.7%) and approaching the SFT-based GUI foundation model, OS-Atlas-Pro-7B (71.2%). This improvement reflects the synergy between Blink-phase attention and Link-phase symbolic reward, which jointly stabilize execution and reduce accumulated errors in extended interaction sequences. In GUI-Odyssey, a benchmark that emphasizes hierarchical decision-making and interface switching, BTL-UI-7B reaches an SR of 65.4%, significantly surpassing GUI-R1-7B (38.8%). Although there is still a performance gap compared to UI-TARS, the proposed BTL-UI is comparable to large-scale SFT models such as OS-Atlas, while requiring significantly less training data.

Table 4: GUI grounding accuracy on Screenspot-Pro [38]. **Bold** means the best results, and underline means the second best results. Avg. denotes the average performance on all subtasks.

| Model | Method | Model Size | Dev | | Creative | | CAD | | Scientific | | Office | | OS | | Avg. |
|---|---|---|---|---|---|---|---|---|---|---|---|---|---|---|---|
| | | | Text | Icon | Text | Icon | Text | Icon | Text | Icon | Text | Icon | Text | Icon | |
| GPT-4o [39] | ZS | - | 1.3 | 0.0 | 1.0 | 0.0 | 2.0 | 0.0 | 2.1 | 0.0 | 1.1 | 0.0 | 0.0 | 0.0 | 0.8 |
| Qwen2-VL [4] | ZS | 7B | 0.5 | 0.0 | 2.6 | 0.0 | 1.5 | 0.0 | 6.3 | 0.0 | 3.4 | 1.9 | 0.9 | 0.0 | 1.6 |
| Qwen2.5-VL [5] | ZS | 3B | - | - | - | - | - | - | - | - | - | - | - | - | 23.9 |
| Qwen2.5-VL [5] | ZS | 7B | - | - | - | - | - | - | - | - | - | - | - | - | 29.0 |
| OS-Atlas-Base [23] | ZS | 7B | 33.1 | 1.4 | 28.8 | 2.8 | 12.2 | 4.7 | 37.5 | 7.3 | 33.9 | 5.7 | 27.1 | 4.5 | 18.9 |
| ShowUI [35] | SFT | 2B | 16.9 | 1.4 | 9.1 | 0.0 | 2.5 | 0.0 | 13.2 | 7.3 | 15.3 | 7.5 | 10.3 | 2.2 | 7.7 |
| UGround [22] | SFT | 7B | 26.6 | 2.1 | 27.3 | 2.8 | 14.2 | 1.6 | 31.9 | 2.7 | 31.6 | 11.3 | 17.8 | 0.0 | 16.5 |
| UGround-V1 [22] | SFT | 7B | - | - | - | - | - | - | - | - | - | - | - | - | 31.1 |
| SeeClick [7] | SFT | 9.6B | 0.6 | 0.0 | 1.0 | 0.0 | 2.5 | 0.0 | 3.5 | 0.0 | 1.1 | 0.0 | 2.8 | 0.0 | 1.1 |
| CogAgent [6] | SFT | 18B | 14.9 | 0.7 | 9.6 | 0.0 | 7.1 | 3.1 | 22.2 | 1.8 | 13.0 | 0.0 | 5.6 | 0.0 | 7.7 |
| UI-TARS [9] | SFT | 2B | 47.4 | 4.1 | 42.9 | 6.3 | 17.8 | 4.7 | 56.9 | 17.3 | 50.3 | 17.0 | 21.5 | 5.6 | 27.7 |
| UI-TARS [9] | SFT | 7B | **58.4** | **12.4** | **50.0** | 9.1 | 20.8 | 9.4 | **63.9** | **31.8** | 63.3 | 20.8 | 30.8 | 16.9 | **35.7** |
| UI-R1 [30] | RFT | 3B | 11.2 | 6.3 | 22.7 | 4.1 | 27.3 | 3.5 | 42.4 | 11.8 | 32.2 | 11.3 | 13.1 | 4.5 | 17.8 |
| GUI-R1 [10] | RFT | 3B | 26.4 | 7.8 | 33.8 | 4.8 | **40.9** | 5.6 | 61.8 | 17.3 | 53.6 | 17.0 | 28.1 | 5.6 | - |
| GUI-R1 [10] | RFT | 7B | 23.9 | 6.3 | 49.4 | 4.8 | 38.9 | 8.4 | 55.6 | 11.8 | 58.7 | **26.4** | 42.1 | 16.9 | - |
| **BTL-UI** | RFT | 3B | 47.4 | 4.8 | 29.8 | 11.9 | 28.9 | 7.8 | 44.4 | 14.5 | 48.6 | 11.3 | 32.7 | 4.4 | 27.1 |
| **BTL-UI** | RFT | 7B | 53.9 | 7.3 | 26.7 | **15.9** | 35.9 | **14.6** | 47.2 | 13.0 | 62.7 | 24.7 | 55.7 | 19.7 | 33.7 |

Table 5: GUI planning accuracy on AndroidControl [36] and GUI-Odyssey [37]. **Bold** means the best results, and underline means the second best results.

| Model | Method | Model Size | AndroidControl-Low | | | AndroidControl-High | | | GUI-Odyssey | | |
|---|---|---|---|---|---|---|---|---|---|---|---|
| | | | Type | GR | SR | Type | GR | SR | Type | GR | SR |
| GPT-4o [39] | ZS | - | 74.3 | 38.7 | 28.4 | 63.1 | 30.9 | 21.2 | 37.5 | 14.2 | 5.4 |
| OS-Atlas-Base [23] | ZS | 7B | 73.0 | 73.4 | 50.9 | 57.4 | 54.9 | 29.8 | 60.4 | 39.7 | 27.0 |
| Qwen2.5-VL [5] | ZS | 3B | 62.0 | 74.1 | 59.3 | 47.8 | 46.5 | 38.9 | 37.4 | 26.5 | 26.7 |
| Qwen2.5-VL [5] | ZS | 7B | 83.4 | 87.1 | 62.5 | 68.7 | 59.7 | 47.1 | 55.6 | 37.8 | 34.4 |
| SeeClick [7] | SFT | 9.6B | 93.0 | 73.4 | 75.0 | 82.9 | 62.9 | 59.1 | **71.0** | 52.4 | **53.9** |
| Aria-UI [41] | SFT | 3.9B | – | 87.7 | 67.3 | – | 43.2 | 10.2 | – | **86.8** | 36.5 |
| Aguvis [24] | SFT | 7B | – | – | 80.5 | – | – | 61.5 | – | – | - |
| OS-Atlas-Pro [23] | SFT | 4B | 91.9 | 83.8 | 80.6 | 84.7 | 73.8 | 67.5 | 83.5 | 61.4 | 56.4 |
| OS-Atlas-Pro [23] | SFT | 7B | 93.6 | 88.0 | 85.2 | 85.2 | 78.5 | 71.2 | 84.5 | 67.8 | 62.0 |
| UI-TARS [9] | SFT | 2B | **98.1** | 87.3 | 89.3 | 81.2 | 78.4 | 68.9 | 93.9 | 86.8 | 83.4 |
| UI-TARS [9] | SFT | 7B | 98.0 | **89.3** | **90.8** | 83.7 | **80.5** | **72.5** | **94.6** | **90.1** | **87.0** |
| UI-R1 [30] | RFT | 3B | 79.2 | 82.4 | 66.4 | 57.9 | 55.7 | 45.4 | 52.2 | 34.5 | 32.5 |
| GUI-R1 [10] | RFT | 3B | 83.7 | 81.6 | 64.4 | 58.0 | 56.2 | 46.6 | 54.8 | 41.5 | 41.3 |
| GUI-R1 [10] | RFT | 7B | 85.2 | 84.0 | 66.5 | 71.6 | 65.6 | 51.7 | 65.5 | 43.6 | 38.8 |
| **BTL-UI** | RFT | 3B | 95.6 | 86.1 | 84.8 | 84.0 | 71.4 | 63.4 | 84.4 | 77.2 | 64.0 |
| **BTL-UI** | RFT | 7B | 96.8 | 88.5 | 88.0 | **88.2** | 76.9 | 69.2 | 84.6 | 78.4 | 65.4 |

## 4.3 Ablation Study

As shown in Table 6, to clarify the contributions of each component in our BTL framework, we conduct an ablation study on the AndroidControl-High benchmark. When trained only with SFT, BTL-UI achieves a baseline performance with an SR of 60.6%. While further using the generated Blink data, SFT obtains a 5% improvement. This proves that Blink data is not only suitable for RFT,

Table 6: Ablation study of BTL-UI. All ablation experiments are evaluated on the AndroidControl-High benchmark by evaluating the grounding and planning capabilities of the agent.

(a) Ablation study of training method and BTL. Blink Data refers to the data contribution in §3.2. BTL Reward denotes the reward design in §3.3.

| SFT | RFT | Blink Data | BTL Reward | AndroidControl-High | | |
|---|---|---|---|---|---|---|
| | | | | Type | GR | SR |
| - | - | - | - | 68.7 | 59.7 | 47.1 |
| ✓ | | | | 79.4 | 63.9 | 60.6 |
| ✓ | | ✓ | | 86.4 | 69.9 | 65.6 |
| | ✓ | | | 86.2 | 71.3 | 65.4 |
| | ✓ | ✓ | ✓ | **88.2** | **76.9** | **69.2** |

(b) Ablation study of Blink Phase ROIs.

| $\lambda$ | AndroidControl-High | | |
|---|---|---|---|
| | Type | GR | SR |
| 1 | 87.0 | 72.1 | 66.6 |
| 2 | 87.6 | 72.8 | 67.4 |
| 3 | 88.0 | 74.2 | 68.1 |
| 4 | 86.8 | 75.6 | 68.4 |
| 5 | 88.2 | **76.9** | 69.2 |
| 6 | **89.4** | 73.1 | **69.2** |

but also for SFT. Furthermore, RFT without Blink data achieves an SR of 65.6%. After adopting Blink data and BTL reward, SR is improved by 3.6%.

Moreover, we examine the effect of varying the number of Blink ROIs ($\lambda$): increasing $\lambda$ from 1 to 6 steadily improves success rates from 66.6% to 69.2%, after which gains plateau, suggesting an optimal trade-off between annotation complexity and attention coverage. It is observed that from Table 6, as $\lambda$ increases, the performance is saturated, so the final $\lambda$ is selected as 5.

Overall, the ablation results confirm that each element of the BTL framework—Blink Phase for targeted attention, Think Phase for structured reasoning, and the Link Phase for precise validation—plays a crucial role in achieving competitive performance in GUI interaction tasks.

### 4.4 Visualization

We present the visualization results and qualitative analysis in the appendix.

## 5 Conclusion and Limitations

We propose the BTL framework, an innovative GUI interaction architecture inspired by the biological cognitive paradigm of Blink–Think–Link. This framework simulates the human closed-loop system of visual perception, cognitive decision-making, and action execution during GUI operations, overcoming the limitations of traditional outcome-driven RFT approaches. Experimental results show that the BTL-UI agent, developed under this framework, achieves significant performance improvements across a variety of GUI interaction tasks.

We believe that the BTL framework proposed in this study establishes a promising and generalizable paradigm for developing digital assistants that are more natural, efficient, and aligned with human cognition. It not only benefits human-GUI interaction but can also be extended to other human-computer interaction tasks, such as embodied intelligence.

**Limitations**. The proposed BTL framework introduces *<blink>* tag outputs compared to conventional Think-Answer structured outputs. Although the blink-generated ROI regions are adaptive and can be empty (zero-length), they typically increase the output sequence length in most cases. While demonstrating performance improvements across various GUI task metrics, this design incurs additional computational processing overhead.

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

# A Prultmpt

**System Prompt**

You are a GUI Agent capable of reasoning based on user instructions, action history, and the current screenshot. You should first observe the layout of the screenshot and extract $N$ elements RELATED TO the user instruction, where $0 <= N <= 5$. Next, think about the reasoning process BASED ON the observations and instructions in your mind, and then provide the user with the answer.

The observation process (can be None if $N == 0$), reasoning process and answer are enclosed within <blink></blink>, <think></think> and <link></link> tags, respectively, i.e.,

<blink>
<element><id>1</id><bbox>[x0, y0, x1, y1]</bbox><caption>dynamic</caption></element>
<element><id>2</id><bbox>[x2, y2, x3, y3]</bbox><caption>static</caption></element>
<element><id>3</id>.....</element>
<element><id>4</id>.....</element>
</blink>
<think> reasoning process here </think>
<link> answer(["Plan": ..., "Action": "function": ..., ...]) </link>.

where captions must be one of [dynamic, static], "dynamic" refers to the interactive area, and "static" refers to the non-interactive areas, such as text and diagrams in the screenshot.
And the observation can be <blink> None </blink>, if $N == 0$.

**User Instruction Prompt**

You are given a task and your action history, with screenshots. You need to perform the next action to complete the task. You MUST CHOOSE the next action from the following defined action space.
## Action Space
Action 1: Back
- format: {'function': 'Back'}
- purpose: back to the previous screen.
Action 2: Home
- format: {'function': 'Home'}
- purpose: navigate to the home page.
Action 3: Tap
- format: {'function': 'Tap', 'position': [x, y]}
- purpose: tap the specified position.
Action 4: Type
- format: {'function': 'Type', 'text': 'str'}
- purpose: enter specified text at the designated location.
Action 5: Swipe
- format: {'function': 'Swipe', 'direction': 'str'}
- purpose: swipe on the screen in the specified direction.
Action 6: LongPress
- format: {'function': 'LongPress', 'position': [x, y]}
- purpose: long press the specified position
## User Instruction
$High - Level\ Instruction$
## Action History
Step 1: ......
Step 2: ......
## Screenshots
<image>

Table 7: The prompt for the BTL-UI.

The prompt of BTL-UI is shown in Table 7. The system prompt is used to format the output of the model according to the three-phase paradigm of Blink-Think-Link. Moreover, the model outputs according to the format of the system prompt, which is convenient for the calculation of the BTL

reward to adjust the model distribution. As shown in equation 6, because the output of Blink Phase can be ∅, we emphasize that the Blink Phase can be output *None* in the system prompt.

OS-Atlas [23] has found that blindly mixing data from different sources for multitask fine-tuning can significantly harm performance due to action space conflicts. We unify the action space of GUI-Odyssey [37] and AndroidControl [36]. And we prompt the model to select the corresponding action from the defined action space. Furthermore, we declare the format and purpose in the user instruction prompt, so that the model can better understand each action type. For the grounding and high-level tasks, only the high-level instruction will be fed into the model. For the low-level tasks, both the high-level and the low-level instructions will be given to the model. In the end, the interaction history will be added to the user instruction prompt.

# B   Visualization

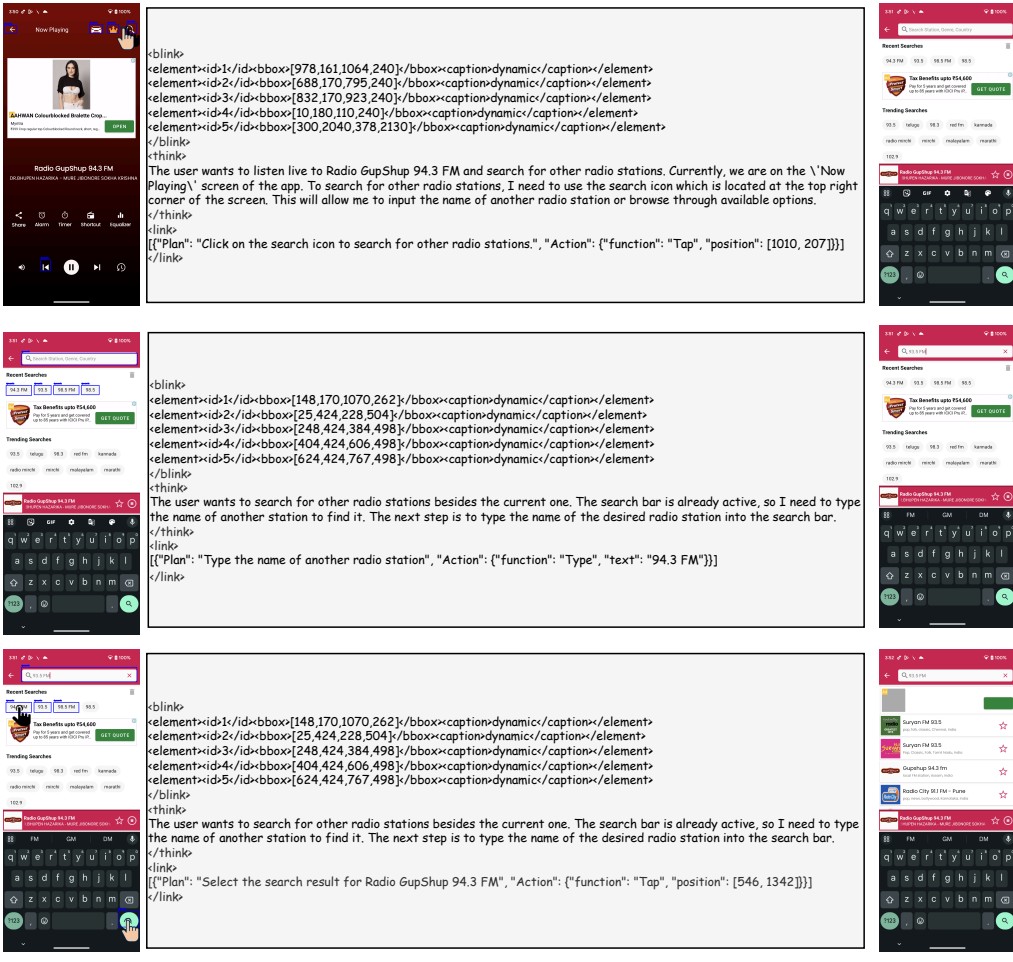

Figure 4: Visualization of the interaction trajectory of the proposed BTL-UI on AndroidControl-High. The corresponding ID of this random case is 19477. And the high-level instruction is 'Listen live to Radio GupShup 94.3 FM and search for other radio stations.' The tap icon in black is the prediction of BTL-UI, and the other is the ground-truth.

The visualization of the interaction trajectory of our BTL-UI is shown in Figure 4. The high-level instruction is "Listen live to Radio GupShup 94.3 FM and search for other radio stations." The Blink Phase can locate the ROIs related to the instruction. And the thinking Phase can reason based on the instruction, interaction history, and candidate area. As shown in step 2 of the interaction trajectory in

Figure 4, in the Blink Phase, BTL-UI not only locates the input box to complete the task, but also analyzes the historical search records in the screenshots.

However, since AndroidControl is an offline interaction benchmark, there are some unreasonable labeling data. For instance, step 2 needs to input the text of "94.3 FM" according to the task instruction. But the search box in the screenshot after interaction shows "93.5 FM", which may affect subsequent interactions. In step 3, the labeled action is to click the search icon. And the search icon is also located in the Blink Phase. Due to the interaction errors in step 2 caused by data noise, the Think Phase of BTL-UI believes that clicking on the "94.3 FM" in historical search records in the screenshot is more reasonable. Therefore, we suppose our BTL-UI has stronger reasoning and error correction abilities.

## C    Experiment Statistical Significance

In this section, we report the experiment's statistical significance. The random factor that affects our results is the sampling of the training process. As shown in Table 1, the training data of our BTL-UI is sampled from various datasets. In the data sampling process, we fix the random seed to 2025 to maintain reproducibility. And the sampled data is further adopted to generate Blink Data, following the pipeline in §3.2. Moreover, BTL-UI adopts the ms-swift [34] framework for RL training. During the training process, we also fix the random seed to 2025 to maintain reproducibility.

