# OpenReview forum: "BTL-UI: Blink-Think-Link Reasoning Model for GUI Agent"
_NeurIPS.cc/2025/Conference — NeurIPS 2025 poster_

### Official Review · Reviewer_RGD1 · 2025-06-23

**Clarity:** 3
**Significance:** 2
**Originality:** 2
**Rating:** 4
**Confidence:** 4

**Summary:**

This paper proposes a new training framework for GUI agents, inspired by biological cognition. The framework simulates the human recognition process through three phases: blink, think, and link. In the blink phase, the agent quickly localizes relevant objects or regions. In the think phase, it formulates a plan to solve the given task. Finally, in the link phase, it generates a precise execution command. In addition to introducing this framework, the authors propose a method for generating training samples specifically for the blink phase. Using both the constructed dataset and existing training data, the authors fine-tune Qwen2.5-VL-3B/7B models with GRPO. The resulting models outperform several baselines on grounding and planning tasks.

**Questions:**

- Could you clarify how the blink module differs from standard attention mechanisms? It seems to perform a similar filtering function. A more detailed discussion may help to better justify the biological inspiration claim.

- Since recent VLMs may already possess strong filtering abilities as the authors utilize Qwen2.5-VL-32B to build the dataset, what additional benefits does your biologically-inspired blink phase provide?

- UI-TARS appears to be a relevant and strong baseline in this domain. Including such results could strengthen the paper. Also, I am wondering why there are gaps in performance scores between the two papers.

**Ethical Concerns:**

["NO or VERY MINOR ethics concerns only"]

**Final Justification:**

After reading the authors' rebuttal, I now have a better understanding of the paper's strengths and its contributions to the community, and I have accordingly raised my score.

However, I strongly recommend that the authors include a comparison with UI-TARS in the paper, as it is a strong baseline and readers would likely be interested in such a comparison. Although the proposed method's overall performance is inferior to that of UI-TARS, its contribution remains valuable given its lower training cost. Additionally, releasing the training dataset would further enhance the utility and reproducibility of the work.

**Limitations:**

Yes

**Quality:**

2

**Strengths And Weaknesses:**

Quality and Significance
- Leveraging concepts from biological cognition is a reasonable and promising direction for enhancing model performance.

- Although this paper claims to be inspired by biological cognition, it is unclear whether the proposed method meaningfully embodies such principles. First, in the "blink" phase, the method aims to extract more relevant objects from an input image. However, this function appears analogous to the attention mechanism already present in existing VLMs. If the authors equate biological cognition with this kind of filtering process, it may overlap with what attention mechanisms already accomplish. Second, while the authors utilize Qwen2.5-VL 32B to collect the training dataset for the blink, we may argue that such ability is already equipped in recent models if Qwen2.5-VL 32B extract relevant elements correctly; if it is not, then the reliability of the collected dataset is questionable. While the method does achieve improved performance, the gains might stem primarily from the new training set—which appears to use the same domain as in the test set—rather than from any principled use of a biologically-inspired cognitive framework.

- The performance comparisons should be strengthened. UI-TARS is a strong and relevant baseline, as it provides a medium-sized model (7B), pretrained weights, and demonstrates strong performance on grounding and GUI-related tasks. However, this paper does not compare against UI-TARS. Furthermore, UI-TARS uses several of the same baselines (e.g., SeeClick, OS-Atlas, Aguvis) and benchmarks (ScreenSpot, AndroidControl) as this paper. Notably, I observed discrepancies in performance scores between the two—for example, OS-Atlas-Base achieves 50.9 SR on AndroidControl-Low in this paper, compared to 85.2 SR in UI-TARS. It would be helpful to clarify the cause of this gap (apologies if I have misunderstood something).

Clarity
- The paper is generally easy to follow.

- Given the performance discrepancies noted above, it would be beneficial to provide more detailed descriptions of the experimental settings and evaluation procedures.


Originality

- The level of originality appears limited, as many prior studies have drawn inspiration from biological systems (albeit not specifically in the context of GUI agents). Furthermore, as discussed above, it is questionable whether the proposed method genuinely reflects the principles of biological cognition.

---

> ### Author Rebuttal · Authors · 2025-07-31
>
> Dear Reviewer RGD1,
>
> Thank you for your valuable suggestions. We sincerely appreciate your insights. We arrange our responses in the order your questions are raised, and hope they clarify your concerns.
>
> ## Response to W1
> We appreciate the reviewer's question regarding the selection of RFT dataset samples. As found in previous work [1], RFT requires higher data quality than SFT. We employ instruction action checks and blink calibration to select high-quality data for Blink data.
>
> ### 1. Is the Blink Phase Redundant with Attention in VLMs?
> While existing VLMs use attention mechanisms to weight visual features of each patch, the Blink Phase explicitly models the biological dynamics of human visual attention. Specifically, saccadic eye movements rapidly locate task-relevant regions in discrete intervals. Unlike implicit attention weights in VLMs, Blink generates structured ROI annotations (`<blink>` tags) that mimic the "spotlight" of human visual focus. Moreover, previous approaches [2,3] have proven that Chain-of-Thought (CoT) can elicit reasoning in LLM. CoT demonstrates a strong ability to solve complex problems by explicitly outputting the reasoning process step by step, rather than directly answering. In dynamic GUI interaction tasks, BTL-UI mirrors how human visual attention combines stimulus salience and goal relevance by locating several key elements related to the user instruction and outputs them to promote further deep reasoning in the Think Phase. As shown in the ablation study (Table 5-a), if the model output is in the format of `<think>...</think>/n<link>...</link>` without Blink Phase, the success rate (SR) decreased by 3.8%. Therefore, we argue that the Blink Phase serves a distinct role that is not simply equivalent to built-in attention in existing VLMs.
> ### 2. If Qwen2.5-VL-32B Can Extract Elements, Why Not Use It Directly?
> This is a fair and insightful question. We clarify our rationale below:
> - We don't adopt Qwen2.5-VL-32B extract elements. Instead,  in the first stage of Blink data generation, we extract candidate UI elements and their bounding boxes and types using OmniParser, which is an off-the-shelf rule-based UI parser.
> - In the second stage, we use Qwen2.5-VL-32B as the analysis model, filtering and prioritizing UI elements from the first stage, based on task instructions and interaction history, to generate ROI annotations. Then, only data samples that can pass the blink calibration are selected as high-quality data. Specifically, each sample will generate several candidate elements with bboxes during the Blink data generation process. At the same time, the coordinates of the click and long-press actions are annotated in the original dataset. We check if the coordinates of the annotated click are within the bboxes.
>
> In summary, we don't adopt Qwen2.5-VL-32B extract elements directly; instead, we produce high-quality data through a pipeline of elemental extraction, analysis, and verification. We will clarify this more explicitly in Section 3.2 and Figure 3 of the revision.
>
> ### 3. Do Performance Gains Come Merely from In-Domain Data?
> We suppose the performance gains stem from the BTL framework, not just domain overlap. Recent studies [1] suggest that RFT only requires a small amount of high-quality data to stimulate the model's capabilities. And the goal of RFT is to stimulate the model's capabilities rather than inject new knowledge [4,5,6]. Therefore, the quality of training data is more important than the domain. Moreover, as we discussed in the first point and the comparison with the recent RFT-based GUI Agent on various benchmarks, BTL-UI achieves more competitive performance, which is due to the BTL framework, inspired by cognitive studies.
>
> ## Response to W2
> We thank the reviewer for pointing out the absence of direct comparison with UI-TARS and for highlighting the potential discrepancies in baseline numbers.
>
> ### 1. Comparison with UI-TARS
> We acknowledge that UI-TARS is a relevant baseline, as it focuses on GUI interaction and shares benchmarks like ScreenSpot and AndroidControl. Our initial experimental comparison did not include UI-TARS due to oversight. We will add a comparison and detailed analysis with high-performing baselines, including UI-TARS and UGround-v1. A brief comparison is shown below:
>
> - Screenspot: As disclosed in their technical report, UI-TARS is a strong GUI Agent developed based on Qwen2-VL with training on 50B tokens. While our BTL-UI is developed based on Qwen2.5-VL, with RFT on 4K samples.  Although the overall grounding performance of BTL-UI is still lower than that of UI-TARS, it shows certain advantages on the subtasks of Mobile and Web.
> |Method|Size|MT|MI|DT|DI|WT|WI|Avg|
> |-|-|-|-|-|-|-|-|-|
> |UI-TARS|2B|95.2|79.1|90.7|68.6|87.2|78.3|84.7|
> |BTL-UI|3B|96.3|77.3|88.2|57.9|80.0|68.9|80.0|
> |UI-TARS|7B|94.5|**85.2**|**95.9**|**85.7**|90.0|83.5|**89.5**|
> |BTL-UI|7B|**97.1**|83.8|90.2|70.7|88.7|**84.5**|87.2|
>
> - AndroidControl: As disclosed in their official repository, UI-TARS adopts the recent five images as input on multi-step interaction tasks for both training and inference, which is computationally expensive. However, in recent methods, a single image training and inference are adopted. Therefore, to ensure comparability, we reproduce the performance of UI-TARS based on the released checkpoint and prompt (*). Although the overall performance of BTL-UI is still lower than that of UI-TARS, under the same experimental settings (single image), BTL-UI is superior to UI-TARS.
> |Method|Size|Images|High|
> |---|---|---|---|
> |UI-TARS|2B|5|68.9|
> |UI-TARS*|3B|1|53.6|
> |BTL-UI|3B|1|63.4|
> |UI-TARS|7B|5|**72.5**|
> |UI-TARS*|7B|1|58.2|
> |BTL-UI|7B|1|69.2|
>
> ### 2. Clarification on OS-Atlas Score
> Thanks for giving us the opportunity to clarify the performance issues regarding OS-Atlas.
> - OS-Atlas is a GUI foundational model developed based on InternVL-2 and Qwen2-VL with Continuous Pre-Training (CPT) on GUI-specific tasks. OS-Atlas-Base (50.9 on AndroidControl-Low) is the model with CPT and without additional Supervised Fine-Tuning (SFT). Thus, as shown in 'Method' in Table 4, the experimental results of OS-Atlas-Base are zero-shot (ZS).
> - Moreover, the results of OS-Atlas (85.2) in UI-TARS have been fine-tuned on the training set of AndroidControl.
>
> For a more comprehensive comparison and to enhance the readability of the article, we will add experimental results of OS-Atlas-SFT to reduce misunderstandings. We hope our response clears things up for you.
>
> ## Response to W3
> Thanks for your thoughtful observation. While biological inspiration is not new in AI, our BTL-UI distinguishes itself through two unique dimensions:
> 1. Novelty in Domain-Specific Adaptation: Most prior biologically inspired methods [7] target general vision/language tasks. In contrast, we are the first to adapt biological visual cognition to GUI interaction, a domain with distinct challenges: dynamic interfaces, multi-modal elements including text, icons, and buttons, and task-dependent relevance. This domain shift is non-trivial, as GUI agents require modeling human-like interaction logic, which demands reinterpreting biological principles for goal-driven, interactive scenarios.
> 2. Mechanistic Depth Beyond Analogy: Our framework does not stop at superficial analogy to attention. Instead, it operationalizes specific biological processes:
>   - Blink Phase mirrors how human visual attention combines stimulus salience and goal relevance by locating several key elements related to the user instruction and outputs them to promote further deep reasoning in the Think Phase.
>   - BTL Reward Integrates process-outcome reward to simulate the “action-evaluation” loop in human cognition, where aided actions trigger re-focusing (Blink) on new ROIs. This mirrors how humans adjust visual attention based on interaction feedback, a mechanism absent in SFT of VLM.
>   - As shown in the ablation study (Table 5-a), if the model output is in the format of `<think>...</think>/n<link>...</link>` without Blink Phase, the success rate (SR) decreased by 3.8%.
>
> In summary, our goal is not to replicate human cognition, but to abstract useful behavioral priors in GUI interaction tasks to improve the agent's visual perception and reasoning capabilities to enhance generalization.
> ## Response to Q1
> Thanks for raising this important question again. We have discussed this '*Response to W1*'. Please kindly refer to it.
> ## Response to Q2
> Thanks for raising this insightful question again. We have discussed this '*Response to W1*' and '*Response to W3*'. Please kindly refer to them.
>
> ## Response to Q3
> Thanks for highlighting the relevance of UI-TARS as a strong baseline in the GUI agent domain again. We have discussed this '*Response to W2*'. Please kindly refer to it.
>
> [1] Liu Z, Sun Z, Zang Y, et al. Visual-rft: Visual reinforcement fine-tuning[J]. arXiv preprint arXiv:2503.01785, 2025.
>
> [2] Wei J, Wang X, Schuurmans D, et al. Chain-of-thought prompting elicits reasoning in large language models[C]. Advances in Neural Information Processing Systems, 2022, 35: 24824-24837.
>
> [3] Guo D, Yang D, Zhang H, et al. Deepseek-r1: Incentivizing reasoning capability in llms via reinforcement learning[J]. arXiv preprint arXiv:2501.12948, 2025.
>
> [4] Gandhi K, Chakravarthy A, Singh A, et al. Cognitive behaviors that enable self-improving reasoners, or, four habits of highly effective stars[J]. arXiv preprint arXiv:2503.01307, 2025.
>
> [5] Yue Y, Chen Z, Lu R, et al. Does reinforcement learning really incentivize reasoning capacity in llms beyond the base model?, 2025[J].  arXiv preprint arXiv:2504.13837, 2025.
>
> [6] Shah D J, Rushton P, Singla S, et al. Rethinking reflection in pre-training[J]. arXiv preprint arXiv:2504.04022, 2025.
>
> [7]  Fukushima K. Neocognitron: A self-organizing neural network model for a mechanism of pattern recognition unaffected by shift in position[J]. Biological cybernetics, 1980, 36(4): 193-202.

---

> > ### Comment · Reviewer_RGD1 · 2025-08-04
> > **Response to the authors' rebuttal**
> >
> > ### On Is the Blink Phase Redundant with Attention in VLMs?
> >
> > I agree that the Blink phase contributes to performance improvements, which is unsurprising given that localizing relevant interface elements is crucial for downstream tasks. However, the observed gains appear to result from conventional techniques—namely, the use of a dedicated training set and well-designed reward functions for localization. Therefore, I question whether the claim of introducing an "innovative framework" that simulates human cognition may be overstated.
> >
> >
> > ### On Comparison with UI-TARS
> >
> > How are the average scores in the first table computed? If the six listed numbers are used to calculate the averages, the results should be 89.13 for UI-TARS and 85.83 for the proposed method. If these calculations are correct, the performance improvement over the original Qwen2.5-VL baseline (84.8) is marginal, whereas the gap between the proposed method and UI-TARS is more substantial. This raises concerns about the limited improvement offered by the proposed approach.

---

> ### Author Response · Authors · 2025-08-05
> **Response to Reviewer RGD1**
>
> ### Response to '*On Is the Blink Phase Redundant with Attention in VLMs?*'
> We appreciate the reviewer's concern regarding the potential overstatement of our framework’s novelty. We agree to localizing the relevant interface element itself, which is a well-established technique. However, our intention is not to claim innovation in isolated components, but rather to present the framework as a coherent and cognitively inspired processing pipeline that explicitly models human-like GUI interaction through **structured perception, reasoning, and action** stages, which can be seen in recent studies [1,2,3]. Each phase is individually supervised and functionally distinct, yet collectively optimized to support interpretable and generalizable predictions. We believe this end-to-end design, grounded in cognitive principles and validated through stage-wise ablations, is what sets our approach apart from prior work. We will revise our wording to better reflect this system-level contribution without overstating novelty at the component level.
>
> [1] Li Y, Liu Z, Li Z, et al. Perception, reason, think, and plan: A survey on large multimodal reasoning models[J]. arXiv preprint arXiv:2505.04921, 2025.
>
> [2] Sapkota R, Cao Y, Roumeliotis K I, et al. Vision-language-action models: Concepts, progress, applications and challenges[J]. arXiv preprint arXiv:2505.04769, 2025.
>
> [3] Liu H, Guo D, Cangelosi A. Embodied intelligence: A synergy of morphology, action, perception and learning[J]. ACM Computing Surveys, 2025, 57(7): 1-36.
>
> ### Response to '*On Comparison with UI-TARS*'
> Thanks for raising the question about the calculation of the average scores! We want to emphasize that the average scores in the first table are not the simple average of the six listed numbers, but the overall performance of ScreenSpot. The details of ScreenSot are shown as follows:
> | Task         | Instances |
> | ------------ | --------- |
> | Mobile Text  | 273       |
> | Mobile Icon  | 229       |
> | Desktop Text | 194       |
> | Desktop Icon | 140       |
> | Web Text     | 230       |
> | Web icon     | 206       |
> | Total        | 1272      |
>
> The average score of BTL-UI-7B can be calculated simply:
> $\frac{97.1 \times 273 + 83.8 \times 229 + 90.2 \times 194 + 70.7 \times 140 + 88.7 \times 230 + 84.5 \times 206}{1272} \approx 87.2$
> The same rules apply to the calculation of UI-TARS and Qwen2.5-VL.

---

> > ### Comment · Reviewer_RGD1 · 2025-08-06
> >
> > Thank you for the clarification. I now have a better understanding of the paper's strengths and its contributions to the community, and I have accordingly raised my score.
> >
> > However, I strongly recommend that the authors include a comparison with UI-TARS in the paper, as it is a strong baseline and readers would likely be interested in such a comparison. Although the proposed method's overall performance is inferior to that of UI-TARS, its contribution remains valuable given its lower training cost. Additionally, releasing the training dataset would further enhance the utility and reproducibility of the work.

---

### Official Review · Reviewer_96rE · 2025-07-03

**Clarity:** 3
**Significance:** 3
**Originality:** 3
**Rating:** 5
**Confidence:** 3

**Summary:**

This paper introduces BTL-UI, a new GUI agent framework inspired by human cognitive processes during graphical interface interactions. The authors propose a three-stage model called Blink-Think-Link (BTL): (1) Blink for visually identifying relevant UI regions, (2) Think for reasoning about the task, and (3) Link for producing executable actions.

To support this framework, they design two key components: (1) A Blink Data Generation pipeline that automatically annotates regions of interest using a large vision-language model. and A BTL Reward, which integrates feedback across the three stages—rewarding structure, attention, and action correctness.

The resulting agent, BTL-UI, is trained using a GRPO-based reinforcement fine-tuning method and evaluated on multiple GUI benchmarks for grounding and planning. It achieves state-of-the-art performance across several tasks, surpassing both supervised and prior RFT-based agents.

**Questions:**

Please refer to weakness section. The confusion I had is around the dataset used for RFT, it would be great if the authors can provide more detailed information on the data.

**Ethical Concerns:**

["NO or VERY MINOR ethics concerns only"]

**Final Justification:**

The rebuttal has addressed my concerns. It would be great if the authors can include the experiments with RFT on larger models to further strengthen the paper, but I don't see it as a reason to reject based on my discussion with the authors. Therefore, I increase my score to accept.

**Limitations:**

Yes

**Quality:**

3

**Strengths And Weaknesses:**

**Strengths**
- The paper proposes a new framework, Blink-Think-Link (BTL), inspired by how humans interact with GUIs. This adds an explicit visual attention step (<blink>) before reasoning and acting, which is not investigated in prior GUI agent work.
- The authors design a full pipeline with automated ROI annotation (blink data), a structured output format, and a reward scheme that gives feedback across all three phases. The reward mechanism avoids issues like partial-credit “reward hacking” found in previous RFT works.
- BTL-UI achieves state-of-the-art performance on several GUI benchmarks, showing improvements over strong supervised and RFT-based baselines. Ablations show that each part of the framework helps.


**Weakness**
- It is not clear how the final dataset samples used for RFT are selected from their original dataset. Are they randomly sampled from the original dataset and then go through the Blink data generation pipeline? Or are there criteria to select specific samples from the original dataset which would better fit the Blink data generation pipeline?
- I am not very sure how the first stage of the data generation pipeline is done. are the bounding boxes extracted by an off-the-shelf detection model, or provided by the original dataset?
- Since Qwen2.5-VL-32B is used for the second stage, how good is the data quality after second stage? I imagine the data quality will be bounded by the capability of Qwen2.5-VL-32B. It should be ok when we are finetuning smaller models I guess, but I am curious whether this is a general recipe to improve performance of large models, e.g., whether we can use this data to RFT Qwen2.5-VL-32B itself and have similar performance improvement.

**Minor Weakness**
- Typos
  - L268/269: `BLink data` -> `Blink data`; sometime it is `blink data` or `Blink Data` throughout the text
  - L269" `archives and SR of xxx` -> `achieves an SR of`
  - L268: `Blikn data` -> `Blink data`
- Other presentation suggestions
  - Define the acronyms when they are first introduced. e.g., GRPO

---

> ### Author Rebuttal · Authors · 2025-07-31
>
> Dear Reviewer 96rE,
>
> Thank you for your valuable feedback. We truly value your insights and are fully committed to addressing these points to enhance the quality of our work.
>
> ## Response to W1
> We appreciate the reviewer's question regarding the selection of RFT dataset samples. As found in previous work [1], RFT requires higher data quality than SFT. We adopt instruction action check and blink calibration to select high-quality data for Blink data.
> 1. Instruction Action Check: We found that there are samples in AndroidControl and GUI-Odyssey, where the labeled action doesn't match the instruction. To filter out high-quality samples, we use regular matching to extract keywords (e.g, click, scroll, and swipe) from the annotated actions and instructions, and verify their consistency.
> 2. Blink Calibration: Each sample will generate several candidate elements with bboxes during the Blink data generation process. At the same time, the coordinates of the click and long-press actions are annotated in the original dataset. Then, we check if the coordinates of the annotated click are within the bboxes. Only data samples that can be matched are selected as high-quality data.
>
> Finally, we sample training data from source datasets after instruction action check and blink calibration, following a fixed random seed (2025) to ensure reproducibility. Moreover, we will release the training data to promote further study.
>
> ## Response to W2
> Thanks for seeking clarification regarding how bounding boxes are obtained in the first stage of our Blink data generation pipeline.
>
> To clarify, the bounding boxes used in the Blink Phase are **not provided by the original datasets**, such as AndroidControl and GUI-Odyssey, as most of these datasets lack detailed UI element annotations. Moreover, as we discussed in '*Response to W1*', these datasets only provide the coordinates of the click, but not the bounding boxes. Instead, we extract candidate UI elements and their bounding boxes using an **off-the-shelf rule-based UI parser**, which we refer to as OmniParser.
>
> We will clarify this more explicitly in Section 3.2 and Figure 3 of the revised manuscript. Thank you again for highlighting the need for a clearer explanation of this step.
>
> ## Response to W3
> We appreciate the reviewer’s concern about the data quality from the second stage of the pipeline and its applicability to larger models.
> 1. Regarding data quality: The second stage, which uses Qwen2.5-VL-32B as the analysis model, filters and prioritizes UI elements based on task instructions and interaction history to generate ROI annotations. As we discussed in '*Response to W1*', the data samples that pass the blink calibration process are selected as high-quality data. Moreover, recent studies [1] suggest that RFT only requires a small amount of high-quality data to stimulate the model's capabilities. Therefore, we suppose the capability of Qwen2.5-VL-32B has less influence on the final data quality.
> 2. Regarding applicability to larger models: The mainstream view of recent approaches [2,3,4] is that RFT can further stimulate the model's ability rather than injecting new knowledge into it. From this perspective, the capacity of base models and data quality are the key to determining the upper bound. As discussed in point 1, Qwen2.5-VL-32B is adopted as the analysis model to produce high-quality data, but not the knowledge distillation similar to a teacher-student network. Moreover, GUI-R1 adopted Qwen2.5-VL-7B to rollout high-quality data to RFT Qwen2.5-VL-3B and Qwen2.5-VL-7B, and both the performance improvements are solid. In summary, we believe that using the Blink data to RFT Qwen2.5-VL-32B, we can gain similar performance improvement.
>
> ## Response to W4
> Thanks for your careful reading and helpful feedback on writing clarity and presentation.
> 1. We will correct all spelling errors and inconsistencies, including:
>   - Replacing “BLink data”, “Blink Data”, and “Blikn data” with the consistent term “Blink data” throughout the paper.
>   - Fixing “archives an SR” to the correct phrasing “achieves an SR”.
> 2. We will also ensure that all acronyms (e.g., GRPO: Group Relative Policy Optimization) are properly defined when first introduced in the main text (Section 3.4) and included in the glossary of symbols in the appendix.
>
> These revisions will be reflected in the updated version. We appreciate the reviewer’s attention to detail in helping improve the clarity and professionalism of the paper.
>
> ## Response to Q1
> We appreciate the reviewer’s follow-up request for more detailed information on the dataset used for RFT. We fully agree that a clear understanding of the data construction pipeline is essential for reproducibility and transparency. Regarding the confusion you mentioned, we have already responded point by point. We will clarify this more explicitly in Section 3.2 and Figure 3 of the revision.
>
> [1] Liu Z, Sun Z, Zang Y, et al. Visual-rft: Visual reinforcement fine-tuning[J]. arXiv preprint arXiv:2503.01785, 2025.
>
> [2] Gandhi K, Chakravarthy A, Singh A, et al. Cognitive behaviors that enable self-improving reasoners, or, four habits of highly effective stars[J]. arXiv preprint arXiv:2503.01307, 2025.
>
> [3] Yue Y, Chen Z, Lu R, et al. Does reinforcement learning really incentivize reasoning capacity in llms beyond the base model?, 2025[J].  arXiv preprint arXiv:2504.13837, 2025.
>
> [4] Shah D J, Rushton P, Singla S, et al. Rethinking reflection in pre-training[J]. arXiv preprint arXiv:2504.04022, 2025.

---

> > ### Comment · Reviewer_96rE · 2025-08-05
> >
> > I am a little concerned about the statement that the authors made regarding to applicability to larger models.  I appreciate the authors pointing out the related studies on the upper bound of RFT. These studies are all on LLMs. As far as I know, there is not yet a comprehensive study on the upper bound of RFT on VLMs. Whether or not the conclusion on LLMs transfer to VLMs remain unclear. However, I don't think this falls under the scope of this paper. But I am happy to discuss with the authors.
> >
> > In addition, my concern is exactly around data quality bounded by Qwen2.5VL-32B as the analysis model.  Basically, the data quality especially the ROI annotations would rely on how good the analysis model is. It is great that there is additional process (the blink calibration process) to ensure data quality. However, it is still unclear to me whether using Qwen2.5-VL-32B in the second stage would somehow limit the data quality for RFT on Qwen2.5-VL-32B itself.
> >
> > Again, I don't see this as a reason to reject. But would love to learn more from the authors.

---

> > > ### Author Response · Authors · 2025-08-06
> > > **Response to Reviewer 96rE**
> > >
> > > We thank the reviewer for the thoughtful comments and are glad to engage in this discussion.
> > > ***
> > > ### Response to '*the upper bound of RFT on VLMs*'
> > > We agree with the reviewer that comprehensive studies on the upper bound of RFT for VLMs are scarce, and most conclusions are drawn from LLMs. Subjectively speaking, our intuitive feeling is that the relevant conclusions can be transferred to the VLM domain, and the main reasons are as follows:
> > > - **Parameter**: The structure of mainstream VLM consists of a visual encoder (around 600M), an adapter (almost negligible parameters) and a corresponding LLM (2B/7B/72B,...). From a parameter perspective, the upper bound of VLMs' capabilities is primarily determined by the upper bound of their LLM. Thus, we suppose that most of the conclusions about LLM are applicable to VLM.
> > > - **Training Paradigm**: It is widely recognized that the pre-training stage, especially when exposed to long-chain Chain-of-Thought (CoT) data, is crucial for embedding latent reasoning abilities in LLMs. Post-training methods like SFT and RFT do not introduce fundamentally new capabilities but instead guide the model to express and utilize what it has already learned. When developing VLM based on LLM, Qwen2.5-VL [1] , Kimi-VL [2] , and MiMo-VL [3] incorporate Long CoT data in both the pre-training and post-training stages, aiming to align the model's visual capabilities and activate its reasoning capability.
> > > - **Modeling**: CoT reasoning mainly depends on the LLM’s ability to produce coherent multi-step reasoning processes in language. In a VLM, the visual encoder simply converts images into token-level embeddings; the actual reasoning happens entirely within the LLM. Since the visual encoder is shallow, it adds little to the reasoning depth. Therefore, the model’s reasoning limit is determined by the LLM’s capacity to perform abstract inference over tokenized inputs. Just like in pure LLMs with CoT prompting, the reasoning ceiling in VLMs is similarly limited by the underlying LLM’s abilities.
> > >
> > > Nevertheless, we acknowledge that the absolute upper bound of VLMs in the RFT setting remains an open question. We agree it would be valuable for future work to investigate this in the VLM context systematically, and we thank the reviewer for highlighting this important direction.
> > >
> > > ### Response to '*data quality bounded by Qwen2.5-VL-32B*'
> > > We appreciate the reviewer’s concern. Due to the blink calibration process, we suppose that the introduction of Qwen2.5-Vl-32B as an analysis model may affect the difficulty distribution of the data rather than the quality of the data. As an analytical model, due to the model's capability boundary, the output may be different from the annotation and thus be filtered out. Therefore, the difficulty of the training data finally constructed is relatively moderate or easy. When RFT Qwen2.5-VL-32B itself, the data difficulty may be slightly easier, which may produce little signal and reduce learning efficiency [4], which may require generating more data for training.
> > >
> > > > Moderate Difficulty. The RL prompt-set should be neither too easy nor too hard, both of which may produce little signal and reduce learning efficiency. We assess the difficulty of each problem using the SFT model’s pass@k accuracy and select only problems with moderate difficulty.
> > > >
> > > > ——KIMI K2 technical report
> > >
> > > In future work, we will explore how data quality and difficulty affect the performance and generalizability of GUI agents.
> > >
> > >
> > > [1] Bai S, Chen K, Liu X, et al. Qwen2. 5-vl technical report[J]. arXiv preprint arXiv:2502.13923, 2025.
> > >
> > > [2] Team K, Du A, Yin B, et al. Kimi-vl technical report[J]. arXiv preprint arXiv:2504.07491, 2025.
> > >
> > > [3] LLM-Core-Team Xiaomi. Mimo-vl technical report[J]. arXiv preprint arXiv:2506..03569, 2025., 2025.
> > >
> > > [4] Team K, Bai Y, Bao Y, et al. Kimi K2: Open Agentic Intelligence[J]. arXiv preprint arXiv:2507.20534, 2025.

---

> > > > ### Comment · Reviewer_96rE · 2025-08-08
> > > >
> > > > Thanks again for the detailed response! I really appreciate the authors' continued efforts and openness during the discussion.
> > > >
> > > > I would agree to our different opinions regarding the `upper bound of RFT on VLMs`, especially whether `post-training methods introduce fundamentally new capabilities` or not. As far as I know, the long-horizon agentic capabilities (such as computer using) are mostly learned through post-training, and it is hard to harvest such capabilities directly from pre-training. I would argue that the current definition of pre-training w.r.t VLMs is not as clean cut as LLMs. e.g., the use of long CoT data during pre-training of VLMs, in my humble opinion, rather feels like a SFT stage for adapting LLMs to multimodal scenario. The differences between SFT and pre-training in the existing VLM literature is mostly the data scale and data quality.
> > > >
> > > > Regarding the `data quality bounded by Qwen2.5-VL-32B`, I thank the authors again for the open discussion. It would be interesting to experiment with RFT on Qwen2.4-VL-32B, with the data filtered with Qwen2.4-VL-32B as the analysis model, to see whether the data difficulty would bound the learning efficiency. And my hutch is that it may also further bound the final performance due to learning efficiency.
> > > >
> > > > With all that said, I am happy with the rebuttal, and will raise the score to accept.

---

> ### Author Response · Authors · 2025-08-09
> **Response to Reviewer 96rE**
>
> Dear Reviewer 96rE
>
> Thanks for your constructive question and strong support. We really appreciate how carefully you considered our responses and your willingness to reconsider the paper's score. Your thoughtful feedback has been incredibly helpful in refining our research. We will continue working to conduct further experiments to enhance our work based on your valuable feedback.
>
> Best regards,
>
> Authors

---

### Official Review · Reviewer_NRM3 · 2025-07-06

**Clarity:** 3
**Significance:** 3
**Originality:** 3
**Rating:** 5
**Confidence:** 4

**Summary:**

This paper introduces BTL-UI, a GUI agent framework based on the "Blink-Think-Link" (BTL) paradigm that models human cognitive processes during graphical user interface interactions by decomposing tasks into three sequential stages - Blink (visual attention and element localization using <blink> tags), Think (reasoning with <think> tags), and Link (action execution with <link> tags).

The framework includes two novel contributions: 1) an automated Blink data generation pipeline that creates region-of-interest annotations using a two-stage parsing and filtering process, and 2) a BTL reward mechanism that provides process-outcome integrated supervision through dual format rewards, blink rewards for ROI accuracy, and link rewards for action correctness.

BTL-UI demonstrates state-of-the-art performance across multiple GUI benchmarks including ScreenSpot, AndroidControl, and GUI-Odyssey.

**Questions:**

Apart from the questions naturally raised in the weaknesses part, I would love to discuss more on the fundamental innovation of the paper.
I might consider the core "innovation" of stage-wise processing can be more accurately characterized as a regularization technique designed to improve data fitting and behavioral control. If so, the imposed stage structure may fundamentally limit the framework's extensibility:
- Information Transmission Loss: Inter-stage dependencies create bottlenecks where critical information may be lost or inefficiently transmitted between processing phases.
- Local Optimization Constraints: Stage-wise optimization may prevent the system from achieving global optima that end-to-end learning approaches could discover.
- Task Decomposition Rigidity: The fixed Blink-Think-Link structure may prove inadequate for tasks requiring different decomposition strategies or parallel processing of multiple information streams.

How do you consider the proposed solution useful in the long run, especially if the benchmark/application becomes more complex?

**Ethical Concerns:**

["NO or VERY MINOR ethics concerns only"]

**Final Justification:**

The authors have addressed key concerns regarding latency, real-time optimization, and scalability, providing evidence of the framework's potential for practical deployment. While the framework shows promising results, further testing in real-world environments and more flexible task decomposition strategies are needed, but current version is enough for a good academic paper. Overall, the paper’s technical contribution outweighs the limitations, justifying a borderline accept rating.

**Limitations:**

yes

**Quality:**

3

**Strengths And Weaknesses:**

Strength
- The cognitive-inspired "Blink-Think-Link" paradigm provides an interpretable framework that aligns GUI interaction with human cognitive processes, offering valuable insights into structured task decomposition.
- Strong empirical performance across multiple benchmarks (ScreenSpot, AndroidControl, GUI-Odyssey) with consistent SOTA results and thorough ablation studies validating each framework component.
- Well-designed reward mechanism integrating process and outcome supervision, automated ROI generation pipeline, and efficient GRPO optimization reducing memory overhead compared to traditional RL approaches.

Weakness
- As the authors acknowledge, the three-stage processing pipeline introduces significant latency that may compromise real-time GUI interaction requirements, potentially limiting practical deployment scenarios. However, modern GUI applications require sub-millisecond response times, and the additional processing stages could fundamentally compromise user experience in real-world deployments. Therefore, some missing points from the current paper could be considered as weaknesses:
    - Latency Benchmarks: No metrics or experiments analyzing inference or interaction latency in practical real-world settings.
    - Real-Time Optimization Strategies: No discussion of methods to mitigate computational demands (e.g., employing lightweight models, approximate inference techniques, or moving toward end-to-end systems that avoid intermediate steps).
    - Scalability Testing: The absence of deployment experiments on consumer-grade hardware or devices limits insight into the GUI agent’s performance outside high-performance computing environments.

- The experiments focus on structured tasks such as ScreenSpot and AndroidControl, primarily involving short-term interactive behaviors (clicking, positioning, swiping and other static tasks). For complex nested tasks (such as multi-step planning and non-linear task objective adjustments), it is not clear whether the reward mechanism or policy model can effectively handle critical intermediate steps. The reward design in this paper emphasizes "process reward accuracy" (such as Blink relative correct regions), but these rewards still lean toward explicit short-term task objectives and are not suitable for complex tasks that require continuous re-derivation of long-term goals.

- The paper lacks crucial implementation details that compromise reproducibility: Missing hyperparameter specifications (learning rates, reward function weights, optimization schedules). Not sure these details would be released.

---

> ### Author Rebuttal · Authors · 2025-07-31
>
> Dear Reviewer NRM3,
>
> Thank you for your valuable suggestions. We sincerely appreciate your insights. We arrange our responses in the order your questions are raised, and hope they clarify your concerns.
>
> ## Response to W1
> We sincerely thank you for raising this important point regarding latency, practical real-time optimization, and scalability, especially in the context of real-world GUI applications requiring fast and reliable response times. We respond to each sub-point in detail below:
> 1. Latency Benchmarks: As discussed in the Conclusion and Limitations (Section 5), while our current work focuses on performance metrics like grounding accuracy and task success rate, we recognize the need for explicit latency analysis. We will supplement this by quantifying end-to-end latency across the BTL pipeline. A brief comparison is shown in the following:
> |Model|Size|Format|Tokens|Latency/ms| |
> |-|-|-|-|-|-|
> | | | | | bf16 (H100)|fp8 + medusa (4090)|
> |BTL|7B|Link|58.4|1231|412|
> |BTL|7B|Think + Link|127.7|2425|864|
> |BTL|7B|Blink + Think + Link|185.1|4247|1462|
>
> As shown in the table, after quantizing the model and using Medusa [1] to accelerate it, the BTL can achieve real-time response. In future work, we will aim to decrease the number of output tokens to achieve real-time optimization.
>
> 2. Real-Time Optimization: As discussed in the first point, model quantization and Medusa, which is a simple LLM Inference acceleration framework with multiple decoding heads, can achieve real-time optimization. Moreover, as shown in our ablation study (Table 5-b), the BTL framework supports adaptive $\lambda$ scaling to reduce the number of ROIs in the Blink Phase, which can balance performance and efficiency.
>
> 3. Scalability Testing: Beyond H100 experiments, we have tested on consumer hardware, specifically the RTX 4090, with quantized 7B or real-time interaction, ensuring the framework adapts to constrained resources.
>
> ## Response to W2
> Thanks for this important observation. We would like to clarify that while our experiments include structured benchmarks, they explicitly evaluate multi-step planning and long-term goal adaptation, with design choices in the BTL framework tailored to address such scenarios.
> 1. The BTL Reward mechanism is designed to support continuous re-derivation of long-term goals through its process-oriented supervision. The Blink Reward provides fine-grained guidance at each step by evaluating ROI relevance to the current sub-goal, while the Link Reward ensures alignment with the global objective. And our ablation study (Table 5-a) confirms that the integration of process and outcome rewards is critical for complex tasks. Removing Blink Data or BTL Reward leads to significant drops in AndroidControl-High SR (from 69.2% to 65.6% or 65.4%), indicating that process supervision (e.g., Blink’s ROI guidance) is essential for maintaining alignment with long-term goals during multi-step execution.
> 2. BTL-UI inherently supports GUI multi-trajectory prediction through its GRPO-based optimization framework, enabling it to explore and evaluate multiple potential interaction paths for complex tasks. Specifically, during policy optimization, the model generates a group of completions ($\{o_1,o_2,...,o_n\}$) for each GUI task instance,  each representing a distinct interaction trajectory spanning Blink, Think, and Link Phases. This multi-trajectory generation allows the model to consider diverse strategies—such as alternative ROI selections in the Blink phase or varying reasoning steps in the Think phase—before determining the optimal path.
> In summary, while our current benchmarks include structured tasks, their design incorporates complex nested scenarios, and the BTL framework, via its process-outcome reward integration, demonstrates effectiveness in handling intermediate steps and adapting to long-term objectives.
>
> ## Response to W3
> Thanks for pointing out this important issue regarding reproducibility. We acknowledge that the initial submission omitted several key implementation details due to space constraints. In the revised version, we will include a comprehensive appendix section detailing the following:
> |Hyperparameter|Value|
> |-|-|
> |learning_rate|from 1e-6 to 0|
> |reward function weights|[0.1, 0.3, 0.6]|
> |temperature|0.9|
> |num_generations|8|
> |num_epochs|3|
> |per_device_train_batch_size|1|
> |gradient_accumulation_steps|2|
> |$\beta$(KL coefficient)|0.04|
> |$\epsilon$(clipping parameter)|0.2|
>
> We will clearly state this in the revised Implementation Details(Section 4.1). Moreover, we will release the training code, data, and corresponding checkpoints to ensure reproducibility.
>
> ## Response to Q1
> Thank you for this profound observation, which prompts us to clarify the fundamental nature of the BTL framework and its distinction from mere regularization. We contend that the Blink-Think-Link structure is not merely a regularization technique but a biologically inspired cognitive paradigm that aligns AI-driven GUI interaction with human cognitive mechanisms, thereby addressing the limitations of existing approaches. We address your concerns as follows:
> 1. Information Transmission Loss: The BTL framework is designed to minimize inter-stage information bottlenecks through deliberate architectural choices.  First, the Blink Phase's ROI predictions (`<blink>` tags) are explicitly fed into the Think Phase as structured inputs, ensuring that visual attention signals directly guide reasoning. Second, the BTL Reward mechanism ensures consistency across stages: the Dual Format Reward verifies structural coherence, while Blink and Link Rewards jointly monitor visual grounding and action execution, preventing information loss through multi-stage alignment. Our ablation study (Table 5) shows that the absence of Blink Data or BTL Reward leads to significant performance drops (e.g., SR decreases from 69.2% to 65.6% without Blink Data), confirming that inter-stage information transmission is both efficient and critical.
> 2. Local Optimization Constraints: BTL-UI avoids local optima by globally optimizing the entire pipeline. The GRPO-based policy update (Equation 9) maximizes a joint objective that integrates rewards from all stages, ensuring Blink, Think, and Link Phases are co-optimized rather than tuned independently. The BTL Reward’s process-outcome integration (combining Blink’s granular reward and Link’s outcome feedback) further guides the model toward global Optimization. For instance, in the high-level task of AndroidControl, only the full BTL framework (with all stages and rewards) achieves the highest SR (69.2%), outperforming ablated versions that optimize stages in isolation. This demonstrates that the framework escapes local optima by leveraging cross-stage dependencies.
> 3. Task Decomposition Rigidity: We appreciate the reviewer's insightful comments. Indeed, the fixed Blink-Think-Link decomposition may not be suitable for all GUI tasks (e.g., highly reactive or parallel-perception settings). However, the current structure is parameterized: for example, when no relevant ROI exists, the Blink Phase can output `<blink> None </blink>`, effectively collapsing to Think-Link. Moreover, the BTL framework is inherently extensible and can support adaptive or task-specific decompositions. Specifically, the Blink Phase can be conditioned on the instruction context to modulate its granularity of attention, focusing broadly or narrowly depending on whether the query is high-level or low-level.
> In summary, BTL’s innovation lies in its cognitive alignment with human-GUI interaction, using structured phases to mimic biological processes rather than imposing arbitrary regularization. This design not only improves performance but also enhances interpretability and alignment with human behavior, addressing critical gaps in existing GUI agents.
>
> ## Response to Q2
> We appreciate the reviewer’s forward-looking question regarding the long-term applicability of our BTL-UI framework.
> We believe our approach is particularly well-suited for scaling to more complex benchmarks and real-world applications for the following reasons:
> 1. The core Blink-Think-Link paradigm mirrors the human cognitive process of visual attention, reasoning, and action, an architecture that naturally adapts to more complex GUI environments. As interfaces grow more intricate, such as nested menus and multi-app workflows, the Blink Phase can dynamically locate task-relevant ROIs. This ensures the framework does not rely on fixed element patterns but instead generalizes to novel, complex layouts.
> 2. The GRPO-based optimization supports multi-trajectory exploration, enabling the model to evaluate diverse interaction paths (e.g., alternative ROI selections or reasoning steps) for complex tasks. This capability—generating and comparing multiple completions to maximize long-term advantage—allows BTL-UI to adapt to unexpected changes in complex applications.
> Therefore, we suppose the BTL framework remains useful as benchmarks and applications grow more complex, with empirical results validating its scalability across diverse task types.
>
> [1] Cai T, Li Y, Geng Z, et al. Medusa: Simple llm inference acceleration framework with multiple decoding heads, 2024[J]. arXiv preprint arXiv: 2401.10774, 2024.

---

> > ### Comment · Reviewer_NRM3 · 2025-08-06
> >
> > The author addresses my concern.

---

> > > ### Comment · Reviewer_NRM3 · 2025-08-06
> > >
> > > The author should include their response in the rebuttal to strengthen the paper.

---

> > > > ### Author Response · Authors · 2025-08-06
> > > > **Response to Reviewer NRM3**
> > > >
> > > > Dear Reviewer NRM3
> > > >
> > > > Thanks so much for taking a closer look at our manuscript and rebuttal. Your thoughtful feedback has been incredibly helpful in refining our research. We will continue working and include the content of the rebuttal to further enhance our work.
> > > >
> > > > Best regards,
> > > >
> > > > Authors

---

### Official Review · Reviewer_gfqS · 2025-07-11

**Clarity:** 3
**Significance:** 3
**Originality:** 3
**Rating:** 4
**Confidence:** 4

**Summary:**

1. This paper proposes a new structured output mechanism for RFT-training of GUI agents which uses \<blink\> tags for proposing likely ROIs in images, \<think\> tags for reasoning, and \<link\> tag for grounded action generation.

2. The Blink data generation pipeline produces ROI annotations for training using Omniparser for proposals and Qwen-32B model for ranking.

3. The authors propose a three stage BTL Reward

    1. Dual format reward for Template and content matching

    2. A novel blink reward that rewards the agent based on the overlap with generated ROIs with ROIs generated from blink data annotations. They use the Hungarian algorithm for allocation of matches.

    3. Link reward for action outcome evaluation which exactly matches the full action as well as ground truth coordinate

**Questions:**

1. Is this method’s overall accuracy upper bounded by the proposals that are generated by omniparser? What if the true ROIs do not get generated by Omniparser? Can this method still generate such ROIs as it's trained to favor (reward) the ones generated by Omniparser?

2. Please add the missing baselines mentioned and compare BTL-UI on Screenspot Pro.

3. The exact reward at the link stage might overfit to the exact actions found in the offline datasets, and the agent might not adapt and generalize to real-world use cases. Could you provide a justification and, preferably, experiments in online GUI environments?

4. Please explicitly note and clarify what x, y, and i are in the section on Blink reward; this will make it easier to follow your explanations. Also, include a brief description of what the Hungarian matcher is for better clarity.

**Ethical Concerns:**

["NO or VERY MINOR ethics concerns only"]

**Final Justification:**

The authors added the requested baselines and addressed my key questions. The authors have added the lack of tests on online environments in the limitations section. However, I think explicitly testing in online environments is necessary, which is why I would like to maintain my score of 4 and not raise it further.

**Limitations:**

Yes.

**Paper Formatting Concerns:**

None.

**Quality:**

2

**Strengths And Weaknesses:**

**Strengths**

1. The structured output template is well motivated and generates GUI actions in a principled manner.

2. The writing and method are clear and easy to understand.

3. Clear ablation studies demonstrating the impact of Blink Data and BTL reward.


**Weaknesses**

1. The authors claim SOTA results on multiple benchmarks, but some necessary, high-performing baselines are missing:

    1. No comparison with baselines like UI-Tars [1], UGround-v1 [2] on Screenspot

    2. Similarly, missing comparison with UI-TARS on the AndroidControl benchmark.

2. Missing evaluation on Screenspot Pro [3], which is a more challenging and recent test for GUI agents featuring diverse applications, high-res screenshots, densely packed realistic UI elements, and more complex scenes. This is a standard evaluation benchmark for GUI agents focusing on better grounding.

3. Lack of evaluation in online environments, to demonstrate that the agent does not overfit to offline actions from the training datasets.

**References**

1. Qin, Y., Ye, Y., Fang, J., Wang, H., Liang, S., Tian, S., … Shi, G. (2025). UI-TARS: Pioneering Automated GUI Interaction with Native Agents (arXiv:2501.12326). arXiv.
2. Gou, Boyu, Ruohan Wang, Boyuan Zheng, Yanan Xie, Cheng Chang, Yiheng Shu, Huan Sun, and Yu Su. Navigating the Digital World as Humans Do: Universal Visual Grounding for GUI Agents. arXiv preprint arXiv:2410.05243 (2025).
3. Li, Kaixin, Ziyang Meng, Hongzhan Lin, Ziyang Luo, Yuchen Tian, Jing Ma, Zhiyong Huang, and Tat-Seng Chua. ScreenSpot-Pro: GUI Grounding for Professional High-Resolution Computer Use. arXiv preprint arXiv:2504.07981 (2025).

---

> ### Author Rebuttal · Authors · 2025-07-31
>
> Dear Reviewer gfqS,
>
> Thank you for your valuable feedback. We sincerely appreciate your insights and are fully committed to addressing these points to improve the quality of our work.
>
> ## Response to W1
> Thank you for pointing out the need for a more comprehensive baseline comparison.  Our initial experimental comparison did not include UI-TARS due to oversight. We will add a comparison and detailed analysis with high-performing baselines, including UI-TARS and UGround-v.  We address each of the concerns below:
> - Screenspot: As disclosed in their technical report, UI-TARS is a strong GUI Agent developed based on Qwen2-VL with training on 50B tokens. While our BTL-UI is developed based on Qwen2.5-VL, with RFT on 4K samples.  Although the overall grounding performance of BTL-UI is still lower than that of UI-TARS, it shows certain advantages on the subtasks of Mobile and Web.
>
> |Method|Size|MT|MI|DT|DI|WT|WI|Avg|
> |-|-|-|-|-|-|-|-|-|
> |UI-TARS|2B|95.2|79.1|90.7|68.6|87.2|78.3|84.7|
> |UGround-v1|2B|89.4|72.0|88.7|65.7|81.3|68.9|77.7|
> |BTL-UI|3B|96.3|77.3|88.2|57.9|80.0|68.9|80.0|
> |UI-TARS|7B|94.5|**85.2**|**95.9**|**85.7**|90.0|83.5|**89.5**|
> |UGround-v1|7B|93.0|79.9|93.8|76.4|**90.9**|84.0|86.3|
> |BTL-UI|7B|**97.1**|83.8|90.2|70.7|88.7|**84.5**|87.2|
>
> - AndroidControl: As disclosed in their official repository, UI-TARS adopts the recent five images as input on multi-step interaction tasks for both training and inference, which is computationally expensive. However, in recent methods, a single image training and inference are adopted. Therefore, to ensure comparability, we reproduce the performance of UI-TARS based on the released checkpoint and prompt (*). Although the overall performance of BTL-UI is still lower than that of UI-TARS, under the same experimental settings (single image), BTL-UI is superior to UI-TARS.
>
> |Method|Size|Images|High|
> |---|---|---|---|
> |UI-TARS|2B|5|68.9|
> |UI-TARS*|3B|1|53.6|
> |BTL-UI|3B|1|63.4|
> |UI-TARS|7B|5|**72.5**|
> |UI-TARS*|7B|1|58.2|
> |BTL-UI|7B|1|69.2|
> - Why don't we use UI-TARS as the baseline? When we conducted experiments on UI-TARS, we found that it has a weak ability to follow instructions, as it cannot output in the specified format according to the designed prompt, but instead adheres to its original output format. Furthermore, BTL-UI requires deep reasoning in the Think Phase. We suppose the space for deep reasoning of UI-TARS with large-scale data SFT and DPO is relatively limited. Therefore, we finally developed BTL-UI based on a foundational model Qwen2.5-VL.
>
> ## Response to W2
> Thanks for highlighting the importance of the Screenspot-Pro benchmark. We would like to clarify that we have already evaluated our BTL-UI model on ScreenSpot-Pro, and the corresponding results are provided in Table 6 of the Appendix (Section A). Specifically, compared to existing baselines, our BTL-UI achieves SOTA performance on this benchmark across both 3B and 7B model scales. Furthermore, a brief comparison with UI-TARS is shown in the following:
>
> |Method|Model|Acc|
> |-|-|-|
> |UI-TARS|2B|27.7|
> |BTL-UI|3B|27.1|
> |UI-TARS|7B|35.7|
> |BTL-UI|7B|33.7|
>
> We appreciate the opportunity to emphasize this result in the main paper.  In the revised version, we will highlight the ScreenSpot-Pro results more clearly in the main body (e.g., by merging them into Table 2 or Table 3) and discuss our model’s performance and robustness in this more challenging setting.
>
> ## Response to  W3
> We appreciate the reviewer’s concern regarding the agent's generalization capability beyond offline datasets. Indeed, evaluating GUI agents in online interactive environments is critical for demonstrating robustness and avoiding overfitting to dataset-specific trajectories.
> While our current evaluation focuses on standardized offline benchmarks (e.g, Screenspot series, AndroidControl, and GUI-Odyssey) adopted by recent methods, we emphasize that several components of the BTL-UI framework were explicitly designed to mitigate overfitting:
> - Our Blink Phase guides attention toward semantically relevant UI regions rather than relying on memorized element positions.
> - The BTL Reward introduces the Process Reward Model (PRM), which encourages generalizable decision-making rather than imitating fixed trajectories.
> - The proposed BTL-UI is optimized by Group Reward Policy Optimization (GRPO), a relative advantage-based policy optimization method that does not directly depend on supervision of ground-truth actions, further reducing the risk of overfitting to offline data, and demonstrating stronger generalization.
> We acknowledge, however, that online evaluation is an important next step. In our future work, we will integrate BTL-UI into online testing platforms, such as OSWorld [1] and AndroidWorld [2], to assess real-time adaptability. We will include a discussion of this direction in the revised manuscript’s limitations section.
>
> ## Response to Q1
> Thank you for your insightful question regarding the potential dependency of our model on the proposals generated by OmniParser during the Blink phase.
> We clarify as follows:
> 1. The accuracy of BTL-UI  is not strictly upper bounded by OmniParser-generated proposals. The two-stage Blink Data Generation pipeline (Section 3.2) decouples raw element extraction from task-relevant filtering. While OmniParser extracts initial UI elements, the second stage leverages Qwen2.5-VL-32B to dynamically analyze, filter, and prioritize these elements based on task instructions and interaction history, effectively refining OmniParser’s outputs. Only data samples that can pass the blink calibration ('*Response to W1*' to Reviewer 96rE) are selected as high-quality data for RFT.
> 2. At inference time, BTL-UI is free to predict any region it deems relevant. In particular, the Blink Reward (Eq. 6) awards a non‑zero reward even when the model outputs an empty blink set ($P_i=\varnothing$), provided that either no ground‑truth ROI exists on‑screen or the chosen action belongs to the non‑interactive action set $\mathcal{A}^*$. This design explicitly prevents the model from being forced to conform to OmniParser proposals when they are incomplete.
> 3. Importantly, since the BTL-UI has the ability to understand screenshots and ground elements, the model operates independently, guided by its learned policy from BTL Reward over screenshots and instructions. Furthermore, the visualization (Appendix C) demonstrates that the BTL-UI is capable of discovering novel ROIs and reasoning beyond annotated ground-truth.
> We agree this limitation should be more explicitly discussed, and we will add a clarification in the revision to ensure readers are aware that the OmniParser is used to extract UI elements from data structures but does not bound the model’s capability to discover or reason about novel UI regions.
>
> ## Response to Q2
> Please kindly refer to '*Response to W1*' and '*Response to W2*'.
>
> ## Response to Q3
> We appreciate the reviewer’s concern about the Link Reward’s potential overfitting to offline datasets and its generalization to real-world online environments. Here, we justify the design and provide supporting evidence, along with plans for online validation:
> 1. Justification of the Link Reward Design: The Link Reward’s strict "all-or-nothing" criterion (reward=1 only if both action type and arguments are exactly correct) is intentionally designed to avoid reward hacking and ensure the semantic consistency between the agent’s intent and execution, which is critical for GUI interactions where even minor errors can derail tasks. Due to its focus on structural correctness rather than memorizing dataset-specific patterns, we suppose BTL-UI does not inherently overfit to offline data. In addition, since RFT is not supervised by the ground-truth directly during training like SFT, RFT has a lower risk of overfitting.
> 2. Planned Online Environment Experiments: Our current experiments on offline benchmarks adopted by recent methods, including AndroidControl and GUI-Odyeesy, that mimic online variability. Moreover, BTL-UI inherently supports GUI multi-trajectory prediction through its GRPO-based optimization framework, enabling it to explore and evaluate multiple potential interaction paths for complex tasks. We suppose these suggest the BTL framework generalizes beyond static offline data, as it handles dynamic state transitions analogous to real-world online interactions. However, due to resource limitations, we are unable to conduct experiments in an online environment further. We will add the absence of online evaluation to the "Limitations" section of the revision and list it as further work.
>
> ## Respone to Q4
> We appreciate the reviewer's suggestion to improve the clarity of our Blink Reward formulation.
> In response, we will revise Section 3.3 to define all variables involved in the reward computation explicitly. Specifically:
> - $i$indexes a given training sample.
> - $x$indexes the predicted region-of-interest (ROI) bounding boxes $p_i^x \in P_i$
> - $y$indexes the ground-truth ROI boxes $g_i^y \in G_i$ in Section 3.2
> We will also revise the definition of index_math in Eq. (5) to clearly reflect these index roles.
> Moreover, we will include a brief inline explanation of the Hungarian matcher:
> ```
> ...we adopt the Hungarian matcher, a classical assignment algorithm used to compute the optimal one-to-one matching between predicted and ground-truth bounding boxes based on IoU scores, under a given threshold τ.
> ```
> [1] Xie T, Zhang D, Chen J, et al. Osworld: Benchmarking multimodal agents for open-ended tasks in real computer environments[J]. Advances in Neural Information Processing Systems, 2024, 37: 52040-52094.
>
> [2] Rawles C, Clinckemaillie S, Chang Y, et al. Androidworld: A dynamic benchmarking environment for autonomous agents[J]. arXiv preprint arXiv:2405.14573, 2024.

---

> > ### Comment · Reviewer_gfqS · 2025-08-07
> >
> > Thank you for the detailed response, adding the comparisons to UI-TARS and UGround-v1, and clarifying the Screenspot-Pro experiments. I will maintain my overall rating as positive feedback.

---

> > > ### Author Response · Authors · 2025-08-08
> > > **Response to Reviewer gfqS**
> > >
> > > Dear reviewer gfqs
> > >
> > > Thank you very much for your kind acknowledgment and for taking the time to review our revised manuscript! We sincerely appreciate your invaluable feedback and support, which have significantly enhanced our work.
> > >
> > > Best regards,
> > >
> > > Authors

---

### Decision · Program_Chairs · 2025-09-17

**Decision:**

Accept (poster)

**Comment:**

This paper proposes BTL-UI, a GUI agent framework built on the “Blink-Think-Link” paradigm, which is inspired by human cognitive processes in graphical interface interactions. The framework decomposes interaction into three phases: Blink for localizing relevant regions, Think for reasoning, and Link for executing actions. To support this design, the authors introduce an automated Blink data generation pipeline and a novel BTL reward scheme that integrates process- and outcome-based supervision. The system demonstrates strong empirical performance across multiple GUI benchmarks, consistently achieving state-of-the-art results.

In general, the paper presents a well-motivated and novel framework that is both biologically inspired and technically sound. The reviewers appreciated its interpretability, clear methodology, well-designed reward system, and strong empirical validation through ablations and benchmark comparisons. The paper is clearly written, easy to follow, and makes a valuable contribution to the development of advanced GUI agents.

However, as pointed out by the reviewers, there are some concerns about the paper. These include limited discussion of latency and real-time deployment constraints, lack of scalability testing on consumer hardware, reliance on Qwen2.5-VL-32B for data quality in Blink annotation, and questions about whether the Blink module truly provides benefits beyond existing attention mechanisms. Additionally, some reviewers noted missing details in experimental setup and implementation, as well as the need for more online environment evaluation. The authors have addressed many of these concerns in the rebuttal, adding baseline comparisons, clarifications on dataset construction, and discussions on scalability and future work.

Overall, the strengths of the paper outweigh its limitations. The work provides a significant and original contribution to the area of GUI agents, backed by thorough experiments and thoughtful design. I recommend accepting the paper.